

# Assessment of GNSS radio occultation refractivity under heavy precipitation

Ramon Padullés[1,2], Estel Cardellach[2], Kuo-Nung Wang[1], Chi O. Ao[1], F. Joseph Turk[1], and Manuel de la Torre-Juárez[1]

[1]Jet Propulsion Laboratory, California Institute of Technology, Pasadena, CA, USA.
[2]Institut de Ciències de l'Espai (IEEC-CSIC), Barcelona, Spain

**Correspondence:** Ramon Padullés (ramon.padulles.rullo@jpl.nasa.gov)

**Abstract.** A positive bias at heights between 3 and 8 km has been observed when comparing the radio occultation retrieved refractivity with that of meteorological analyses and re-analyses, in cases where heavy precipitation is present. The effect of precipitation in RO retrievals has been investigated as a potential cause of the bias, using precipitation measurements interpolated into the actual three dimensional RO raypaths to calculate the excess phase induced by precipitation. The study consisted in comparing the retrievals when such extra delay is removed from the actual measurement and when it is not. The results show how precipitation itself is not the cause of the positive bias. Instead, we show that the positive bias is linked to high specific humidity conditions regardless of precipitation. This study also shows a regional dependence of the bias. Furthermore, different analyses and re-analyses show a disagreement under high specific humidity conditions and in consequence, heavy precipitation.

## 1 Introduction

Radio Occultation (RO) technique uses opportunistic Global Navigation Satellite System (GNSS) signals to sound the atmosphere. The signal trajectory, travelling from GNSS satellites to Low Earth Orbiters (LEO), is bent due to the index of refraction gradients of the atmosphere. Such bending can be inferred using the phase derivative observable (Doppler shift) obtained by dedicated receivers in the LEOs. Under the assumption of a spherically symmetric atmosphere, the bending angle profile can be integrated to a vertical profile of the refractive index, $n(h)$, through Abel inversion (e.g. Kursinski et al., 1997; Hajj et al., 2002).

Refractivity is defined to account for the deviations of the index of refraction from unity, and is related to geophysical parameters by (e.g. Thayer, 1974; Kursinski et al., 1997):

$$N(h) = (n(h) - 1) \times 10^6 = 77.6 \frac{P}{T} + 3.73 \times 10^5 \frac{e}{T^2} -$$
$$40.3 \times 10^6 \frac{n_e}{f^2} + O\left(\frac{1}{f^3}\right) + 1.4 W_w + 0.6 W_i \tag{1}$$

where $P$ is the total pressure (mbar), $T$ is temperature (K), $e$ is the partial water vapour pressure (mbar), $n_e$ is the electron density ($\mathrm{m}^{-3}$), $f$ is the frequency (Hz), and $W_{w,i}$ are the liquid and ice water contents ($\mathrm{g \cdot m^{-3}}$), respectively. These terms are





classified as dry, wet, ionospheric and scattering terms. The dry term is dominant below 60-90 km, while the wet term becomes significant in the lower troposphere. The ionospheric term becomes dominant above 60-90 km, and its leading contribution is removed by a combination of two of the frequencies used by GNSS satellites (L1 = 1.575 GHz; L2 = 1.228 GHz) (Vorob'ev and Krasil'nikova, 1994). The scattering terms (i.e. $W_{w,i}$) are generally much smaller compared to the other refractivity terms in the lower troposphere. Therefore, they are usually neglected in the retrieval of the atmospheric variables, and when RO

measurements are assimilated into the Numerical Weather Prediction (NWP) models.

A commonly used method to retrieve temperature, pressure and water vapour from RO observations is the one dimensional variational retrieval (1DVAR). It consists in obtaining the most probable atmospheric variable combining a priori atmospheric information with the observations in a statistically optimal way (Healy and Eyre, 2000). Usually, these a priori values are obtained from global meteorological analyses or reanalyses. On the other hand, bending angle and refractivity profiles are

directly assimilated into NWP (e.g. Healy et al., 2005; Cucurull et al., 2007), with a high positive impact in the weather forecasts (Cardinali and Healy, 2014).

An unavoidable link exists between NWP models and RO retrieved temperature, pressure and moisture, due to the fact that RO products use a priori information from the models, and models assimilate RO observations. Yet, differences exists between their products, and its understanding is important in order to detect weaknesses and potentially improve the performance of

models.

In this study we compare RO refractivity observations with the global weather analyses and re-analyses, in the presence of precipitation. These analyses have coarse spatial resolution, which has a direct impact in the treatment of heavy precipitation. At these scales, convective processes need to be parametrized. In turn, convective parametrization (CP) has been identified as one of the major source of errors in the modelling of heavy precipitation (e.g. Arakawa, 2004). RO technique offers unique potential

to study the interaction between heavy precipitation and vertical thermodynamic processes within the atmosphere, since their signals can penetrate into thick clouds and their products have high vertical resolution. Recent investigations by Cardellach et al. (2014, 2017) and Padullés et al. (2016) have shown potential to retrieve vertical precipitation information adapting RO receivers to collect polarimetric observables (Pol-RO). Therefore, Pol-RO emerge as a technique that could provide relevant simultaneous information of precipitation and thermodynamics (e.g. moisture), to advance in the understanding of the processes

linking vertical structure of moisture and heavy precipitation.

While such products are not yet available, in this study we investigate precipitation induced features in standard (non-polarimetric) RO products using collocations (i.e. space and time coincidences) between the COSMIC/FORMOSAT-3 mission (Anthes et al., 2008) and the Tropical Rainfall Measurement Mission (TRMM) (Kummerow et al., 2000) and Global Precipitation Measurement (GPM) (Hou et al., 2014) missions, and we compare such features with those of analyses and re-analyses.

The refractivity from analyses and re-analyses is derived using the temperature, pressure and moisture that they provide, and Equation 1.

A clear positive bias in the RO refractivity with respect to that of some weather analyses and re-analyses is observed between 3 and 8 km height when precipitation is present in the surroundings of the observation. Previous studies have noted similar biases, for example Lin et al. (2010); Yang and Zou (2012); Zou et al. (2012); Yang and Zou (2016). These studies linked the



bias with the liquid and ice water content present in the observation site, suggesting that the scattering term from Equation 1 should not be neglected, but used to correct RO refractivity observations instead. However, our approach in this study is different and takes into account the 3-D structure of precipitating medium. Here, the impact of precipitation is assessed directly in the Doppler shift observable, using three dimensional collocations of precipitation structures and realistic RO ray trajectories, together with computational simulations of the effect of the scattering of the propagating signal by liquid and solid water

particles. Afterwards, the causes of the observed bias are discussed with focus on the performance of the used analyses and re-analyses, especially under high specific humidity conditions. The reason to proceed this way is because solely comparing the RO observations with data from analyses and re-analyses, one could not make a clear distinction on whether the bias is due to the observation technique limitations or the weather analyses limitations.

This paper is structured in the following way. The details of the data and collocations used for this study are explained in

section 2. In section 3 the bias in the comparison between RO observations and analyses and re-analyses is introduced. section 4 presents the results of the assessment of the precipitation induced delay into the RO observables. And in section 5 the specific humidity is assessed as the source of the refractivity bias. Finally, section 6 contains a discussion on the results.

## 2   RO, analyses, and precipitation data

The COSMIC/FORMOSAT-3 RO products are obtained from the University Corporation for Atmospheric Research (UCAR)

COSMIC Data Analysis and Archive Center (CDAAC). The observed RO refractivity is obtained from the Level-2 wetPrf products, along with the retrieved temperature, pressure, and water vapour partial pressure at every 0.1 km of altitude, between surface level and 20 km. The observed refractivity included in the wetPrf files is the same product as in the atmPrf files, provided here at the same height levels as the other thermodynamic products. These observations are collocated with the European Center for Medium range Weather Forecast (ECMWF) ERA Interim re-analysis (e.g. Dee et al., 2011), the ECMWF

high resolution operational analysis, and the National Centers for Environmental Prediction (NCEP) operational analysis, the Global Forecast System (GFS) (NOAA/NCEP, 2003). These collocated profiles are obtained also at the CDAAC in the Level-2 eraPrf, echPrf and gfsPrf products, respectively. The RO products are interpolated into the analyses height levels when the comparisons are performed.

Data from the TRMM and GPM precipitation missions are obtained from the NASA Goddard Earth Sciences Data and

Information Services Center (GES DISC). The TRMM data used here is the Level-2 orbital 2B31 products, that provide vertical structure information of precipitation and has a limited swath coverage. The used GPM data is the final run of the Integrated Multi-satellitE Retrievals for GPM (IMERG) products (Huffman et al., 2017), that provide surface rain rate for the region comprised between ±60° latitude. In order to assess the precipitation intensity and structural characteristics, data from the vertically profiling TRMM radar are used, while the GPM IMERG data is used to increase the statistics. The TRMM 2B31

products provide precipitation information for the region sensed by the TRMM Precipitation Radar (PR), such as rain rate, with a swath of approximately 250 km, a horizontal resolution of 5 × 5 km, and a vertical resolution of 250 m. The IMERG product provides an estimate of the global surface precipitation with an horizontal resolution of $0.1° × 0.1°$ and every 30 minutes.





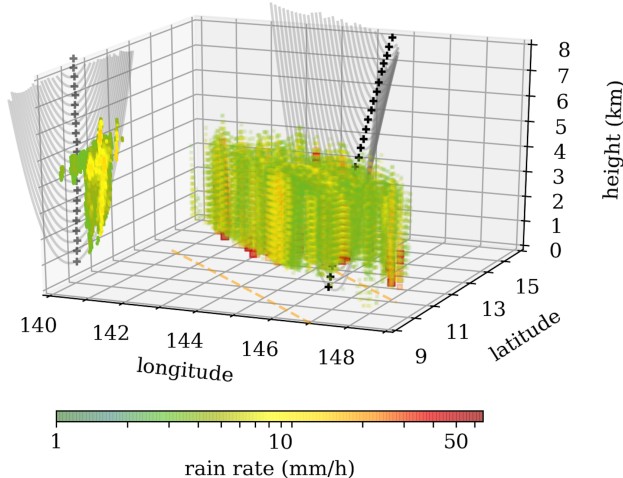

**Figure 1.** Three dimensional collocation of a RO event with a TRMM PR precipitation measurement. It corresponds to the coincidence between the C004.2006.329.22.20.G19 RO event and the 2B31.20061125.51450 TRMM PR product. Here the precipitation structure is shown in a 3 dimensional grid, along with the set of RO ray trajectories (in gray). Black stars indicate the tangent point of the rays. Only a few rays are shown for illustration purposes. The orange dashed lines indicate the edges of the TRMM PR swath. The interpolated precipitation information (rain rate) into the RO plane is shown in the 2-Dimensional projection in the latitude-height plane.

## 2.1 Collocations with the TRMM PR

The COSMIC/FORMOSAT-3 RO products between 2006 and 2015 were compared against TRMM orbital products looking for coincident observations in space (RO soundings within TRMM swath) and time (both observations within ±15 minutes). After 2013, the number of COSMIC/FORMOSAT-3 RO observations dropped significantly. However, the quality and distribution of the observations was not affected. 16,881 collocated events resulted from such comparison. These events were then classified depending on the presence or not of precipitation and its intensity. Henceforth, each event is linked to the number of pixels of the TRMM radar with a reflectivity ($Z$) larger than 30 dBZ, used as a proxy for heavy precipitation events, in the surroundings (100 km) of the occultation location within the radar swath.

For each event with evidences of precipitation in its surroundings, the approximated RO ray trajectories have been simulated using ray-tracing techniques. Therefore, it has been possible to interpolate the precipitation information into the set of RO ray trajectories. An example of such interpolation is shown in Figure 1. We can therefore estimate the amount of precipitation crossed by each of the rays, and compare it with the RO observables such as the excess phase (or the Doppler shift), the Signal to Noise Ratio (SNR) or the atmospheric vertical retrievals. We use this information to assess the impact of precipitation into the RO signal propagation and its retrievals, as described in section 4.



## 2.2 Collocations with GPM IMERG

In order to improve the statistics of collocated profiles we have performed a larger scale collocation using the GPM IMERG products (spatial coverage between ±60 deg and every 30 min) and all the COSMIC/FORMOSAT-3 RO products of 2016. We can greatly expand the number of collocations by considering only the surface precipitation rate. For each of the COSMIC/FORMOSAT-

3 RO events, the corresponding IMERG product has been identified, and the precipitation retrieval has been linked to the RO event. This results in 259,231 RO events from which the surface precipitation in its surroundings has been identified, with a time resolution of ±15min (IMERG data is provided for every 30 min). For each event, the mean rain rate, the maximum rain rate and the number of pixels with non-zero rain rate, in a region of $2° × 2°$, is stored along with the vertical RO profiles of refractivity, temperature, pressure, water vapour pressure, and the corresponding collocated weather analyses and re-analyses

products.

## 3 Refractivity bias

A clear positive refractivity bias is observed between $\sim 3$ and $\sim 8$ km of altitude when precipitation is present in the occultation position, with respect to the refractivity from weather analyses and re-analyses. In Figure 2 the bias is shown, for the comparison between the GPM IMERG collocated RO products and the three different analyses and re-analyses introduced in section 2. In

this case, the data are separated according to the amount of rain in the surroundings: events with no rain (no-rain profiles) and events where $\langle R \rangle > 10$ (mm/h) in the $2° × 2°$ surrounding area.

While the bias is clearly seen for the three analyses and re-analyses used in the comparison, their performance within heavy precipitation is also different. When precipitation is not present close by the RO sounding, the RO refractivity and that of analyses and re-analyses agree (i.e. no significant bias), as well as among themselves.

In Figure 3 we show the regional dependence of the bias, at a height of 6 km. Here, the globe is divided in hexagons with a diameter of approximately 30 degrees, and the events are separated according to their $\langle R \rangle$: $\langle R \rangle = 0$ mm/h; $0 < \langle R \rangle < 5$ mm/h; and $\langle R \rangle > 5$ mm/h. This separation is shown at each different column, while the rows separate the analyses or re-analyses used in the comparison. This figure shows how the positive bias is present globally under heavy precipitation, although is larger in certain regions, and it depends on the analysis in use. Common features for all three re-analyses are, for example, the positive

bias under heavy precipitation that is present in the West Pacific warm pool, the eastern part of the pacific, south Indian ocean, and over South America. These regions are associated to extreme precipitation features (Liu and Zipser, 2015), either to large extension precipitation events or to precipitation systems with a high deep convective core.

Besides the positive bias in the region above an altitude of 4 km, a negative bias is also clearly observed below 3 km, both for the rainy and no-rain events. This bias is not assessed here, since it has already been discussed previously in other studies (e.g.

Ao et al., 2003; Sokolovskiy, 2003; Xie et al., 2006, 2012; Wang et al., 2017). Similarly, other potential sources of bias have been checked, for example, the angle of incidence of the occultation ray to the receiver, with respect to the transmitter position. The larger the angle, the larger the tangent point drift. This implies that the theoretical spherically symmetric atmosphere could depart from a realistic approximation and induce errors in the retrievals (Foelsche et al., 2011). Also, large incident angles





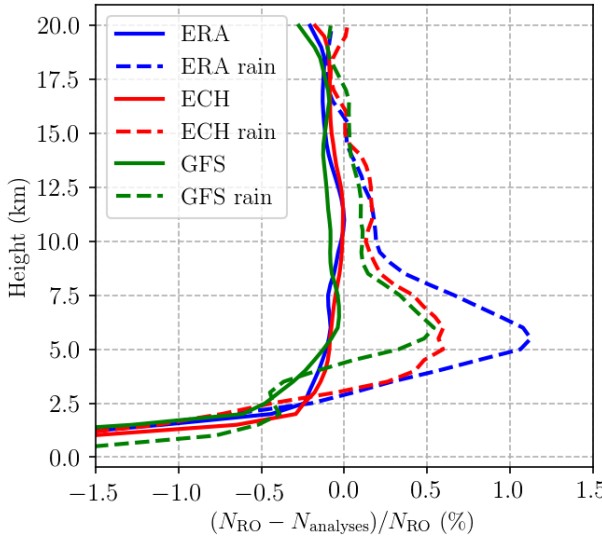

**Figure 2.** Fractional difference between the RO observed refractivity and that from (blue) Era interim re-analysis; (red) ECMWF high resolution analysis; and (green) NCEP GFS operational analysis. The compared profiles are classified between no-rain profiles (solid lines) and heavy rain profiles (dashed lines) according to the GPM IMERG collocations (see subsection 2.2).

correspond to low SNRs, which could be introducing positive biases (Sokolovskiy et al., 2010). Therefore, the positive bias has been checked grouping the occultation events according to its azimuth angle, in addition to rain variables. The results have shown no significant changes for the positive bias, hence the geometry of the observations is excluded as an explanation for the bias.

## 4 Precipitation induced delay

Once other observational known issues are discarded as plausible sources of the bias, the influence of the scattering term in Equation 1 is assessed. In order to further investigate its importance, we have simulated the contribution of the liquid and solid water directly into the excess phase. This is accomplished using 3-Dimensional collocations between the COSMIC/FORMOSAT-3 RO observations and the TRMM PR measurements, which have allowed us to perform realistic simulations of the effects of precipitation in actual RO observables (see Figure 1). This represents a novel approach to the assessment of the positive refractivity bias with respect to previous studies.

The contribution from precipitation on the phase delay of the signal is due to the scattering of the propagating wave by non-spherical raindrops. The delay induced by raindrops (or frozen hydrometeors) with respect to that of free space can be linked to the scattering term of refractivity in Equation 1. For the case in study, the coherent propagation of plane waves is described as the sum of the effects off all the raindrops in a unit volume with various sizes. Formally, the scattered field can be





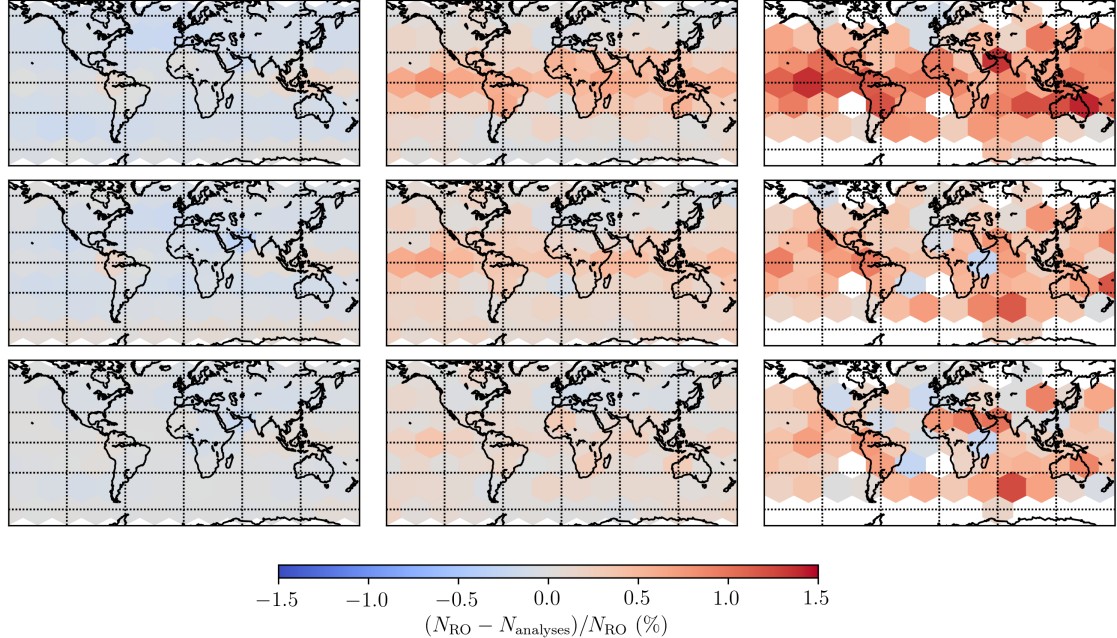

**Figure 3.** Regional averaged fractional difference between the RO observed refractivity and that from (top row) Era interim re-analysis; (middle row) ECMWF high resolution analysis; and (bottom row) NCEP GFS operational analysis; for a height of 6 km. The compared profiles are classified between no-rain profiles (left column; $\langle R \rangle = 0$ mm/h), low and moderate precipitation (middle column; $1 < \langle R \rangle < 5$ mm/h) and heavy rain profiles (right column; $\langle R \rangle > 5$ mm/h). The grid corresponds to hexagons with a diameter of about 30 deg.

expressed as:

$$E^s = T E^i \qquad (2)$$

where $E^i$ is the incident field, $E^s$ is the scattered field, and $T$ is the "transmission matrix" describing the characteristics of the rain medium (Oguchi, 1983). The propagation through rain can be considered as a propagation through an effective medium

5    with two characteristic axes, characterized by the two eigenvalues of $T$, $\lambda_1$ and $\lambda_2$:

$$T = \begin{bmatrix} e^{\lambda_1 l} & 0 \\ 0 & e^{\lambda_2 l} \end{bmatrix} \qquad (3)$$

where $l$ is the propagated distance.

Raindrops fall following gravity and are flattened due to the air drag, becoming approximately oblate-shaped. Here we do not take into account the canting angle effect (raindrops being tilted by wind), for simplicity and because in this situation

10    its effect is secondary. Therefore, $\lambda_{1,2} = -i k_{\text{eff}}^{h,v}$, where the $k_{\text{eff}}$ is the effective propagation constant of the medium, that is complex, and $h$ and $v$ indicate the characteristic axes of the medium (horizontal and vertical).





The effective propagation constant can be expressed as (e.g. Bringi and Chandrasekar, 2001):

$$k_{\text{eff}} = k_0 + \frac{2\pi n_{\text{p}}}{k_0} e_i \boldsymbol{f}(\hat{i},\hat{i}) \tag{4}$$

where $k_0$ is the propagation constant in the homogeneous atmosphere, $n_{\text{p}}$ is the number of particles per unit volume, $e_i$ indicates the unit polarization vector for the linear states, and $\boldsymbol{f}(\hat{i},\hat{i})$ is the scattering amplitude vector in the forward scattering

configuration. The real part of the effective propagation constant induces a phase shift, while the imaginary part induces an attenuation. At L-band, the attenuation due to the scattering by rain can be neglected. The expression of $k_{\text{eff}}$ is defined for a number of identical particles, but can be generalized to a size distribution of particles defined by $N(D)$. Also, the $\boldsymbol{f}(\hat{i},\hat{i})$ can be expressed as the Scattering amplitude matrix, S, using the Jones notation (Jones, 1941). The scattering amplitude matrix (2 × 2) relates the scattered field components to the incident field components in the far field approximation. For a right hand

circularly polarized (RHCP) propagating field, as it correspond to GNSS transmitted signals, a mean effective propagation constant can be defined by:

$$k_{\text{eff}}^{\text{mean}} = \left( \frac{k_{\text{eff}}^h + k_{\text{eff}}^v}{2} \right), \tag{5}$$

hence, the specific phase shift induced only by the raindrops to a circularly polarized incident wave is:

$$\Delta\Phi^{\text{rain}} = \left( \frac{\lambda}{2\pi} \right) \frac{2\pi}{k_0} \int \Re\left\{ \frac{S_{hh}(D) + S_{vv}(D)}{2} \right\} N(D) dD \tag{6}$$

in units of $\text{mm} \cdot \text{km}^{-1}$, where $\lambda$ is the wavelength (mm), $S_{hh,vv}$ are the co-polar components of the forward scattering amplitude matrix in a linear base of polarization, $N(D)$ is the particle size distribution ($\text{mm}^{-1}\text{m}^{-3}$), and $D$ is the diameter of the particles (mm). The forward scattering amplitude matrix is computed for each scatterer, and depends on the scatterer's size, composition, orientation, and shape (see Bringi and Chandrasekar (2001) for a detailed explanation). For this study, the T-matrix code is used in order to compute $S$ for raindrops of all sizes between 0.1 and 8 mm of diameter (Mishchenko et al.,

1996). For the particle shapes, the Beard and Chuang (1987) model is used, which relates the diameter of the each particle with the relationship between its two characteristic dimensions (i.e. its axis ratio). The complex permittivity for liquid water is obtained from Liebe et al. (1991). The $N(D)$ is obtained at each point from the TRMM products, using the same one used to provide rain rate from the TRMM PR reflectivity measurements.

Using the three dimensional collocations we can therefore compute the phase delay that is solely due to precipitation, in the

following way:

- For each collocated event, we have the precipitation information interpolated into the set of RO ray trajectories. The precipitation information (for example, rain rate, water content, etc.), directly or indirectly, is used to infer the $N(D)$ at each point of these trajectories.

- With the $N(D)$, we can compute the specific $\text{d}\Phi^{\text{rain}}$ along each ray using Equation 6, and integrate this quantity along

each ray path:

$$\Phi^{\text{rain}} = \int_L \Delta\Phi^{\text{rain}}(l) dl \tag{7}$$





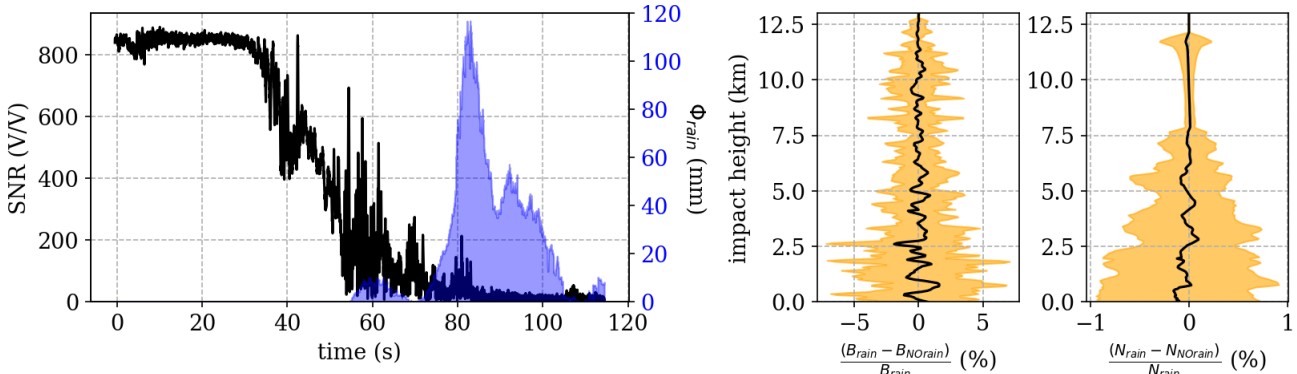

**Figure 4.** (left) Actual SNR (black) corresponding to the RO event C001.2008.345.00.43.G03 (UCAR id), along with the simulated precipitation induced phase delay (blue) as a function of time; (right) Fractional bending angle and refractivity differences between the outputs from the retrieval using the rain-affected profiles and the rain-removed ones, as a function of the impact height. Black lines represent the mean of the 65 cases, while orange shade is the standard deviation.

in units of mm, where $L$ is the ray-path length in km.

For each occultation event that has been 3-d collocated with the TRMM PR, we can have the approximate vertical profiles of precipitation induced delay along with all the currently provided information, such as the total excess phase delay, the strength of the signal, and the retrieved vertical thermodynamic products. In the left panel of Figure 4 we show an example of an occultation actual SNR together with the precipitation induced phase delay.

## 4.1 Precipitation induced phase delay impact

In this section we want to assess the impact that the precipitation induced phase delay has on RO retrievals. To do so we have designed a study that consists in retrieving the bending angle (Phase Matching method (Jensen et al., 2003)) and refractivity (inverse Abel transform (Fjeldbo et al., 1971)) profiles from the total excess phase delay to compare it with the retrieval results when the precipitation induced delays are removed from the original total excess phase. Therefore, the precipitation induced delays obtained in the previous section are removed from the actually observed phase delays, obtaining two profiles called the rain (original) and the rain-free (where the precipitation induced delay has been removed).

The bending angle and refractivity retrieval were attempted on both rain and rain-free excess phases from a total of 65 cases collocated with heavy precipitation events. The bending angle profiles calculated by Phase Matching were smoothed with 200m windows and compared in the same impact height (corresponding impact parameter minus the collocated radius of Earth). Because of the integration nature of inverse Abel transform, the standard deviation (orange shade in right panels of Figure 4) in the retrieved refractivity is much smoother than the one in bending angle profiles. If precipitation had a systematic effect into RO phase delays, a positive bias would be observed in the rain-affected bending angle and refractivity when compared with the rain free bending and refractivity for the same case. However, this effect is absent in the right panels of Figure 4.





The results of nonexistent mean positive bias shown in the right panels of Figure 4 suggest that the positive bias found in the retrieved refractivity compared to the weather analyses and re-analyses is not due to the neglect of the scattering term in the refractivity. Furthermore, it can be seen how on average, the impact of taking / not taking into account the precipitation induced delays when assessing the retrieval increases the variability, implying that the changes of removing precipitation

contribution from the signal propagation can be both positive and negative, rather than only negative. Since the bending angle and refractivity retrieval process depends mostly on the vertical gradient of the excess phase, the precipitation induced excess Doppler, which can be positive or negative, will on average lead to unbiased retrieval results. This extra excess Doppler can be seen as the result of local horizontal inhomogeneity in the refractivity field.

Differently from temperature and pressure, the liquid and ice water content is localized in a small region (compared to the ray

travel distance), and might not be contributing along the whole propagation ray-path of an occultation. Furthermore, the region where liquid and ice water is present might be far from the tangent point. Yet, the refractivity retrieved from a RO observation is located around the RO tangent point, and considered to have an horizontal resolution of about 200 km (e.g. Kursinski et al., 1997). Even though the RO observations are integral quantities, most of the contribution from dry and wet air in the bending angle comes from near the tangent point.

In addition, the RO retrievals rely on the spherical symmetric atmosphere approximation. While it has been proven to work properly for the standard RO thermodynamic products, liquid and solid water content contributions to the excess phase cannot be well captured under such assumption. In consequence, the effect that liquid and solid water content has into the RO retrieved refractivity profiles is to induce small errors such as those characterized in the right panel of Figure 4. Thus, the fact that the scattering terms in Equation 1 are not taken into account is not the cause for the positive refractivity bias observed under heavy

precipitation scenarios.

## 5   Specific humidity as a source of refractivity bias

Once the scattering term in Equation 1 has been discarded as the main source for the refractivity bias, another hypothesis is tested in this section. That is, large scale analyses and re-analyses have problems representing the thermodynamic variables linked to heavy precipitation scenes, in particular high specific humidity conditions.

Using the data described in subsection 2.2, we have studied the refractivity bias as a function of the RO retrieved specific humidity. In turn, these cases are also separated between whether precipitation was present in the surroundings of the observation or not. We have done it for the three analyses/re-analyses: the ERA-interim, the ECMWF high resolution analysis and the GFS, and the results are shown in Figure 5. Revealing results can be found here: the refractivity fractional difference increases with specific humidity, regardless of precipitation. Hence, the refractivity bias is linked to high specific humidity rather than

precipitation itself. However, high specific humidity conditions are strongly correlated with precipitation.

This classification allows us to further investigate four different scenarios: no rain with low specific humidity conditions; no rain with high specific humidity; rain with low specific humidity; and rain with high specific humidity. In this case, the criteria for low and high specific humidity is that the RO retrieved $q$ is lower than 0.5 g/kg and higher than 2.7 g/kg, respectively, in



the cases with no rain, and that the RO retrieved $q$ is lower than 1.0 g/kg and higher than 2.7 g/kg, respectively, in the cases with rain. The $q$ and the fractional refractivity difference are evaluated at a height of 6.5 km. These thresholds are based on the lower and higher 20th and 80th percentiles of data with no rain and rain. For these four classifications, the regional fractional refractivity differences are shown in Figure 6, for the comparison with ERA-interim re-analysis, ECMWF high resolution

analysis and the GFS analysis. Here the globe is divided in hexagons of a diameter of 45 deg.

The results in Figure 6 confirm the results anticipated in Figure 5, i.e. the fractional refractivity bias can be linked to high specific humidity conditions rather than to precipitation itself. From the regional dependence of the fractional refractivity bias some other conclusions can be extracted. The first one is that when there is no rain and the specific humidity is low, the fractional refractivity difference is very small regardless of location and the analyses in use.

The second conclusion one can extract from Figure 6 is that when specific humidity is high, the fractional refractivity difference is positive and reaches large values ($> 1\%$), for all the analyses in use and regardless of the presence of precipitation. In particular, high specific humidity observations are concentrated in the tropics, so the largest positive refractivity bias are in this region, in agreement with Figure 3.

The third conclusion is that precipitation under low specific humidity conditions is rarely observed in the tropics. Under

these conditions, the fractional refractivity difference has a more complicated behavior, more model dependent than the rest of the cases. In this case, no clear positive bias is observed, but a variability depending on the location of the observations. The only situation with a prominent negative bias is observed under this scenario in the west area of Africa.

Finally, even though a positive fractional refractivity difference bias can be linked to high specific humidity conditions, it is also dependent on the analysis in use. For example, the bias is larger in the case of ERA-interim re-analysis, but smaller in the

case of ECMWF high resolution analysis, showing the different performance of models in characterizing precipitation. On the other hand, for no rain and low specific humidity, the performance of the different analyses is similar. The fact that the ERA bias is positive is an indication that models tend to be biased dry, in agreement with Hersbach et al. (2015), which attributed the bias to a problem in assimilating microwave radiance affected by rain.

## 6 Summary and discussion

A systematic positive bias in the fractional refractivity difference has been identified when comparing RO retrieved refractivity with that of weather analysis and re-analysis when heavy precipitation was present in the surroundings of the observation. In this paper, the bias has been shown to be linked to the performance of models under high specific humidity conditions rather than with precipitation itself.

This conclusion has been reached after: (1) assessing the impact of precipitation directly into the RO observables (e.g.

Doppler shift and bending angle), simulating the contribution of realistic three dimensional precipitation structures into the actual RO ray trajectories, and comparing the retrievals after such a contribution is removed; and (2) evaluating the refractivity bias between RO observations and weather analyses under different humidity and precipitation conditions.





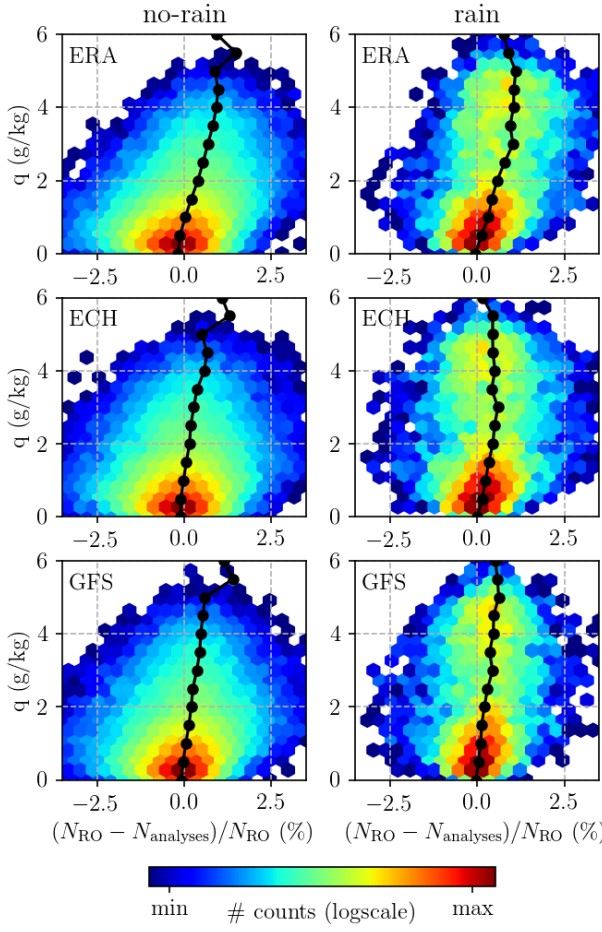

**Figure 5.** Fractional refractivity difference between the observations and analyses as a function of the observed specific humidity at a height of 6 km. The left column corresponds to the no-rain cases and the right column corresponds to the rain cases. The top row shows the results for the comparison of observations and ERA-interim, the middle row show the results for ECMWF high resolution analysis and the bottom row shows the results for the GFS.

First, precipitation has been shown to have little impact on the positive fractional refracitivity bias between RO observations and analyses and re-analyses. Differences in bending angle and refractivity between rain and rain-removed profiles can be both positive and negative, with no clear bias on average (see right panels in Figure 4). If precipitation, through the scattering term in Equation 1, had a systematic positive impact into the RO retrieved refractivity with respect to when precipitation is not present, the study performed in subsection 4.1 would have shown a positive bias as well. Therefore, precipitation itself does not explain such an impact, but the combination of thermodynamic variables associated with heavy precipitation, might. This statement does not mean that precipitation does not enhance the local refractivity where it occurs (which it is), but that such local enhancements do not necessarily lead to a bias.



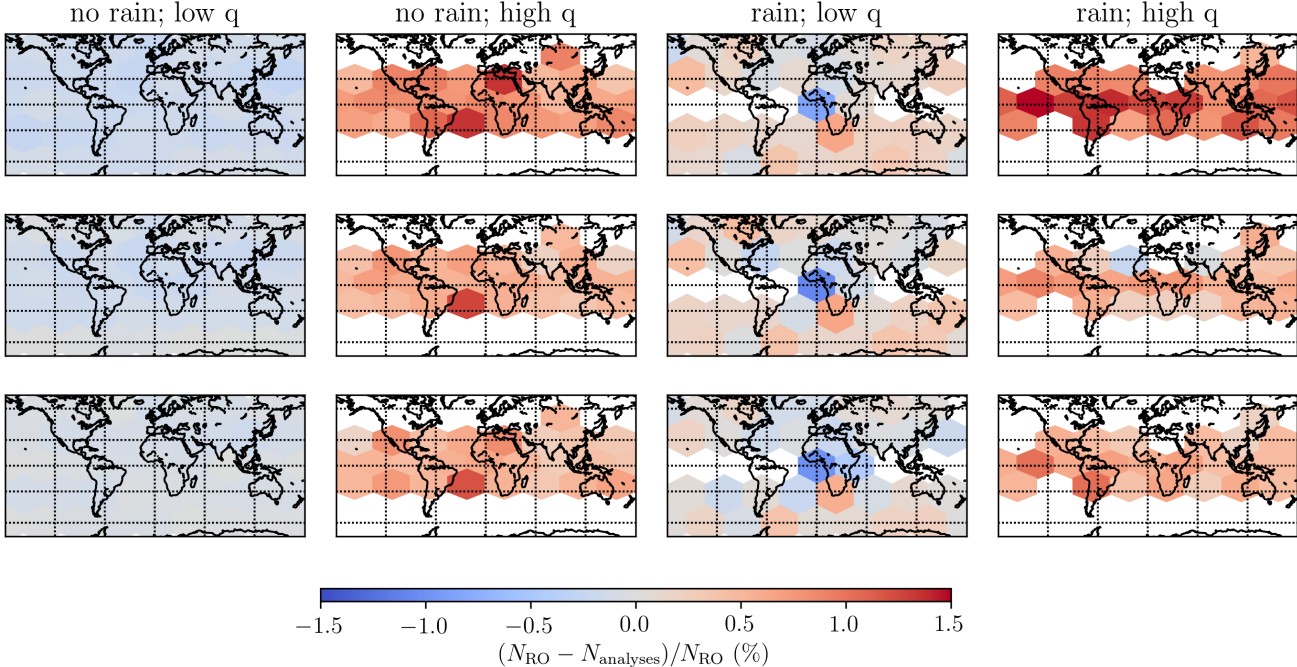

**Figure 6.** Regional averaged differences (colorscale) between the observed and the analysis refractivity. The first row corresponds to ERA interim, the middle row corresponds to ECMWF high resolution and the bottom one to the GFS. The two left rows corresponds to the free of rain data, where in the first row the observed specific humidity at a height of 6.5 km is lower than 0.5 g/kg, and in the second row it is larger than 2.7 g/kg. The two right columns represent rain affected data, where in the third column the observed specific humidity at a height of 6.5 km is lower than 1.0 g/kg, and in the last row it is larger than 2.7 g/kg. The grid here corresponds to hexagons with a diameter of 45 deg.

The fractional refractivity bias between RO observations and analyses has been linked to high specific humidity conditions. The bias appears both in rain and no-rain conditions, and it depends on the analysis and on the geographic region. The fact that the bias is not seen in Figure 2 for the no-rain cases is because most of the no-rain cases have low specific humidity conditions, and they weight much more for the mean value of the fractional refractivity difference. On the other hand, the rain cases have a larger contribution in the high specific humidity region (see Figure 5), contributed mostly by tropical precipitation. This is also seen in the right panels in Figure 6, where precipitation with very high specific humidity conditions is mostly observed around the equator.

The bias in fractional refractivity between observations and analyses implies that the retrieved temperature and moisture will be also biased with respect to models. The positive refractivity bias is associated with a combination of colder retrieved temperature with respect to analyses, and a higher retrieved specific humidity than the one in the analyses. Also, the fact that a difference exists between the different analyses used for this study, and that RO thermodynamic retrievals depend on the model in use, imply that a difference between the retrievals obtained by different processing centers will exist under such conditions if they use different models.




These results stress the need for a better thermodynamic characterization of high specific humidity scenarios, likely to be associated to heavy precipitation. Large scale models are known to have issues with the parameterization of convective processes, hence further investigation in this direction is required. This is the aim of polarimetric radio occultations, which will provide joint products of temperature, pressure, moisture and an indication of the amount of precipitation (mostly sensitive to

5     the heaviest) at each vertical level (Cardellach et al., 2017) with the objective to advance in the understanding of heavy precipitation events, closely linked with high specific humidity conditions. Alternatively, further investigations are being conducted with the aim to make the RO retrievals less dependent on models, which would improve the retrievals itself and provide more independent information of such scenarios.

*Competing interests.* The authors declare that they have no conflict of interest.

10     *Acknowledgements.* Ramon Padulles and K. N. Wang's research was supported by an appointment to the NASA Postdoctoral Program at the Jet Propulsion Laboratory, administered by Universities Space Research Association under contract with NASA. The work conducted at ICE-CSIC/IEEC was supported by the Spanish grant ESP2015-70014-C2-2-R. Part of Cardellach's contribution has been supported by the Radio Occultation Meteorology Satellite Application Facility (ROM SAF) which is a decentralised operational RO processing centre under EUMETSAT. The JPL co-authors acknowledge support from the NASA US Participating Investigator (USPI) program. TRMM and GPM

15     data were obtained courtesy of the NASA Precipitation Processing System (PPS). Part of this work was carried out at the Jet Propulsion Laboratory, California Institute of Technology, under a contract with the National Aeronautics and Space Administration. ©2017. All rights reserved.





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
