# Peer review of "Assessment of GNSS radio occultation refractivity under heavy precipitation"

_Atmospheric Chemistry and Physics, 2018_

## Referee Comment (RC1) · Anonymous Referee #1 · 24 May 2018

Review for Padullés et al., "*Assessment of GNSS radio occultation refractivity under heavy precipitation*"

Summary:

This paper assess the impact of heavy precipitation on GPS radio occultation measurements through comparison of RO profiles within the precipitation and without precipitation as indicated by the satellite radar observations. Systematic positive refractivity errors (or *N*-biases) above ~2.5 km are shown in GPS RO soundings in the presence of precipitation, when comparing to two analyses (ECMWF, GFS) and one reanalysis (ERA-I). The results is consistent with multiple previous studies by Lin et al. (2010), Yang and Zou (2012), Xou et al., (2012) and Yang and Zou (2016). However, these previous studies attribute such positive *N*-bias to the GPS RO sounding retrievals for neglecting the RO refractivity contribution from liquid and ice water contents. In contrast, this paper attributes the *N*-biases to the possible deficiency in global analyses/reanalysis under high specific humidity condition (including both rain and no-rain). The simulation studies were carried out to investigate the contribution of the liquid/ice water content by ray tracing through a 3-D atmosphere with realistic liquid/ice water content estimated from TRMM. The particle size distribution *N*(D) are adapted from the one used in TRMM radar precipitation retrieval. The simulation study of 65 cases confirms that the liquid/ice water content do not introduce significant bias to the RO retrieval. Further analysis of the *N*-bias with respect to the specific humidity shows good correlation with the high specific humidity instead of the precipitation.

Overall, the paper is well written. The science of the paper is significant and it advances our understanding of the liquid/ice water content on the GPS RO measurements, which has puzzled the RO community for quite some time. The discovery of positive *N*-bias (i.e., negative temperature bias, and/or positive moisture bias) likely attributed to the global analyses/reanalysis in the high specific humidity condition will be worth further investigation. Note some more details related to the simulation of liquid/ice water content are needed in the manuscript, and some text and a few figures can be improved.

I would recommend publication of the paper after "minor revision". The comments are listed below:

Major comments:

(1) Section 4 missed some details of the ray-tracing simulation.
   a. The typical range of the liquid and ice water content (along the ray path) derived from TRMM radar reflection measurement (e.g., at different altitude in Fig. 1) should be shown or at least discussed in the manuscript.
   b. The size distribution of the particles, N(D) was not shown.
   c. How typical are those precipitation cases observed by TRMM? For example, the size, rain rate, etc, should be discussed.

(2) The *N*-bias study (e.g., Fig. 6) focused on the high specific humidity based on COSMIC RO wet retrieval. Could the authors use the analyses/reanalysis specific humidity instead, and see whether the *N*-bias pattern will remain the same? Or is there any references that compare the RO specific humidity with the global analyses/reanalyses, which confirm the consistency?

(3) Figure 3:
   a. Add latitude/longitude labels
   b. Add the title for each row with the precipitation rate "<R>"
   c. Maybe put side labels for "ERA-I, ECMWF_An, GFS"
(4) Figure 4:
   a. Why the impact height goes all the way down to 0 km, which will likely be ~2km below the earth surface. Should that be "geometric height" instead, especially for the refractivity error plot? Please verify and make sure it is consistent with the manuscript description in L3 in Section 4.1.
(5) Figure 6:
   a. Add latitude/longitude labels
   b. Add labels after each column title: "no rain low q (<0.5 g/kg)" etc.
   c. Maybe put side labels for "ERA-I, ECMWF_An, GFS"

Technical comments:

The line numbers were messy and not consistent. The following is the best I can do to point to the text in the manuscript.

"liquid and solid water content" → liquid and ice water content

**Page-1 – Sec. 1 (right column)**
L20: remove "of the"

**Page-3 – Sec.2.1**
L01: change to "the global surface precipitation every 30 minutes with a horizontal resolution of 0.1° latitude x 0.1° longitude.

L10: "compared against" → "collocated with" ;
L11: remove "looking for coincident … resulted from such comparison." → "A total of 16,881 COSMIC RO soundings are identified to be within the swath of the TRMM precipitation measurements (250 km), and within +/- 15 minutes."

It's confusing whether the collocation threshold "250 km" (i.e., swath size of TRMM) or "100km" as seen in L19 (Section 2.1)? Please clarify.

Page-3 – Sec.2.2
L03: spatial … between +/-60 deg and every 30 min → every 30 minute with spatial coverage between 60°S and 60°N

L12: remove "(IMERG data is … 30 min)", which is redundant.

**Page-3 – Sec.3**
L10: for the two analyses and one reanalysis

L18: The hexagon with a diameter of ~30 deg is used in Figure 3. What is the sampling looks like? What the minimum/maximum and average sampling number of the collocation within the hexagon?

If the sampling plot will not be shown, it needs to be mentioned/discussed in the text, to justify the choice of "30 deg".

I would expect it is primarily restricted by the sampling, but could it be possible to reduce the size of hexagon and show better spatial pattern?

L22: Revise sentence: "This figure shows the global distribution of the positive refractivity bias under heavy precipitation. The regional difference of the N-bias is evident and the difference among analyses and reanalysis is also shown."

**Page-4**
Figure 2 caption: The compared profiles are …" → The RO profiles are classified into no-rain (solid) and heavy rain (dashed) based on the collocated GPM IMERG precipitation measurements.

The time range should be mentioned in the caption.

**Page-4 – Sec.3**
L10: "The results have shown no …" → "The results (not shown) reveal no significant changes to the positive *N*-bias, and confirms that the RO observation geometry is not a contributing factor to the positive bias.

**Page-4 – Sec.4**
L21: For the case in this study
L22: sum of the effects of all the raindrops
L33: Reference needed for "Raindrops fall following gravity and are flattened … oblate-shape".

L04: What is the subscript 1, 2 refer to? Simply corresponding to "h, and v"? Need to be explained.
L06: that is complex → that is a complex number

**Page-5 – Sec.4**
L35: The REFERENCE is needed for "The N(D) is obtained … TRMM products…"
L36: provide → retrieve

**Page-6 – Sec.4 – Sec. 4.1**

L20: It is a bit odd to see the sentence here. "In the left panel of Figure 4, …induced phase delay."

It should be moved and integrated into the discussion of Figure 4 in Section 4.1, possibly after L04.

Figure 4: I would suggest to use (a, b, c) to identify the three panels, which make it easier to discuss in the manuscript.

Figure 4 deserves more discussion on each panel, especially the Fig. 4a (SNR plot). For example, what is the precipitation rate of the selected case, where, when, and how big is the precipitation feature? How typical is that compare to the other 62 cases?

Could the author(s) also add the excess phase delay without rain to be compared to the result (blue shaded) under heavy precipitation?

L08: effect into RO → effect on RO

L432: "The effect that liquid and solid water content has into the RO … is to induce small errors such as those .. Figure 4." → "the effect of liquid/ice water content on RO refractivity retrieval results **in small errors (??%, need numbers**), which does not introduce obvious biases in both bending angle and refractivity (Figure 4b, c).

L435: "Thus, the fact that …scenarios." → "Thus the scattering terms in Equation 1 should not be the cause for the positive N-bias in the presence of the heavy precipitation."

**Page-6 – Sec.5**

L439 – L457: The two paragraph were not well written and require some revision to improve the points.

Note the author(s) like to use the first person in the manuscript.

L449: "We have done it for the three analyses/reanalyses" → the two analyses and one reanalysis.

L458: "This classification allows us to further .. scenarios:" → "We further classify the collocated COSMIC RO profiles into four different categories:"

L462: the criteria → the threshold

**Page-7 – Sec.5**

L474: I assume that using coarse resolution of "45 deg" instead of "30 deg" was due to the limited sampling? Please offer the description of the map of sampling after separating into four categories. Will the possibly low and un-even sampling affect the results?

L502: "the bias is larger in  ERA-interim …but smaller in  ECMWF … analysis, showing the different performance of model in characterizing precipitation.

Not sure that is a correct interpretation. The physical model used in ERA-I reanalysis and ECMWF analysis should be pretty much the same. The major difference is the spatial resolution as well as the data assimilated. I would argue the resolution might be the major reason behind the difference. Please discuss and justify your reasoning.

---

## Referee Comment (RC2) · Anonymous Referee #3 · 18 Jun 2018

This manuscript investigated the underlying physical cause of a systematic high-bias in retrieved GNSS radio occultation refractivity in the middle troposphere under heavy precipitating situations. Previously people had found similar bias signals but attributed the bias to the scattering caused by raindrop/frozen hydrometers. In this study, the authors interpolated the collocated 3D TRMM-precipitation radar retrieved rain rate to the GPS RO plane and found the difference between simulated SNR and observed ones are not systematically biased. Rather, the authors found some positive relationships between the percentage bias (%) and the collocated specific humidity from three different re-analysis/analyses products. Therefore, the conclusion they made from this work is that the systematic high-bias under heavy precipitation scenes are caused by the corresponded increase in specific humidity.

[Figure]

This idea is novel. The comparisons to multiple observations and reanalysis/analyses products make the conclusion rather solid. The writing is clear and concise. I think this manuscript deserves the final publication in ACP. However, there are some logic caveats I'd like to point out that are either not considered clearly enough when carrying out the data analysis, or not described clear enough to make the readers not confused. There are a few minor glitches that may improve after the revision.

First, with respect to the broad picture of the logic flow: (1) the heavy precipitation scenes are defined by TRMM-PR or IMERG "observations" (by saying observations, I mean retrieved products), but not the precipitation scenes in the reanalysis/analyses products. However, the collocated specific humidity profiles are identified from the re-analysis/analyses. Therefore, when you separate the specific humidity value according to no-rain, light-rain and heavy-rain, it's not necessarily the specific humidity environment for no-rain/light-rain/heavy-rain in the reanalysis/analyses products. I believe this effect is minor as the water vapor field is rather smooth and not as intermittent as the cloud water content field. But I think you need to be clear about this logic difference in the manuscript.

(2) secondly, regarding the collocation and co-incident measurements between GPS-RO and TRMM-PR and IMERG: as precipitation is so transient, especially for heavy precipitation, +/- 15 minutes criterion for co-incident measurements might be too loose. The geo-collocation criterion for TRMM-PR and IMERG was not specified clearly in the context: Do you consider the footprint effect? How do you align the GPS-RO limb-sounding with TRMM-PR type of nadir-viewing instrument?

(3) the simulation of heavy precipitation scenes using the assumed raindrop size and size distribution is more or less questionable to me. As above 5 km melting layer, most of the precipitation-sized particles are frozen hydrometers indeed, and using raindrop assumption throughout the column is not a very good assumption. But once you calculate scattering frozen hydrometers, more free parameters like ice habit, density, etc., will be involved and the uncertainties are huge. I think it is NOT a good idea to COM-

PLETELY exclude the precipitation-sized particle scattering effect out, first because of the aforementioned bullet#1, and also because simulation of scattering effect for any hydrometers, especially when frozen hydrometers are involved, is far from perfect. Actually, since the largest discrepancy occurs at ∼ 5 km in Fig. 1 for all three re-analyses/analyses datasets, I suspect strongly that the extremely complicated situation in the melting layer (at ∼ 5km in the tropics) would at least have certain effect on that.

(4) Since the authors didn't discuss throughout the manuscript that why they reach different conclusions with previous literatures (e.g., Lin et al., 2010; Yang and Zou, 2012; etc. as mentioned in the manuscript), I'm not sure whether it's because the analysis methodology is different? Data sources are different? Assumptions are different? It worths a paragraph or at least a couple of sentences to discuss the differences in your and previous efforts that eventually lead to the discrepancy in conclusions.

(5) As I mentioned in Comment#3 above, this paper didn't explain the vertical structure of the high-bias shown in Fig. 1. Rather, the last two figures (Fig.5 and 6) and related discussions focus on a single altitude (6 km or 6.5 km) and the reason why this altitude is selected was not specified clearly in the text.

Minor points: Page 8, equation (7): please be consistent with dphi or delta_phi. Figure 3 and Figure 6: Since the collocated sample for heavy precipitation scenes is small, it's important to have the statistical significance level shown on the map. Please consider only color statistically significant grids, or overplot the contoured significance level.

Figure 2: Do you have any speculation why ERA-rain looks worse than ECH-rain? Also, please include the standard deviation envelope for each rain curve.

Figure 4: Can you show a case with negative delay of SNR, together with corresponding TRMM-PR rainrate vertical profile projected on the RO plane for the two cases? I don't get in what situation that SNR delay could be reversed.

Regarding Fig. 5 and Fig. 3: if your hypothesis is correct, you should see smaller specific humidity for light-precipitation scenes (Fig. 3, middle column). Is that the case from the collocated re-analysis/analysis?

Page 11, Line 22: I don't understand this statement. Would you please elaborate?

---

## Author Comment (AC1) · 24 Jul 2018

Reviewer #1:

The authors would like to thank the reviewers for spending time and effort reviewing this manuscript. Their comments are deeply appreciated and have resulted in an improved version of the manuscript. Below we address point by point all the comments and recommendations of the reviewers, and at the end of the document there is the new version of the paper with the changes highlighted in blue.

Summary:

This paper assess the impact of heavy precipitation on GPS radio occultation measurements through comparison of RO profiles within the precipitation and without precipitation as indicated by the satellite radar observations. Systematic positive refractivity errors (or N-biases) above ~2.5 km are shown in GPS RO soundings in the presence of precipitation, when comparing to two analyses (ECMWF, GFS) and one reanalysis (ERA-I). The results is consistent with multiple previous studies by Lin et al. (2010), Yang and Zou (2012), Xou et al., (2012) and Yang and Zou (2016). However, these previous studies attribute such positive N-bias to the GPS RO sounding retrievals for neglecting the RO refractivity contribution from liquid and ice water contents. In contrast, this paper attributes the N-biases to the possible deficiency in global analyses/reanalysis under high specific humidity condition (including both rain and no-rain). The simulation studies were carried out to investigate the contribution of the liquid/ice water content by ray tracing through a 3-D atmosphere with realistic liquid/ice water content estimated from TRMM. The particle size distribution N(D) are adapted from the one used in TRMM radar precipitation retrieval. The simulation study of 65 cases confirms that the liquid/ice water content do not introduce significant bias to the RO retrieval. Further analysis of the N-bias with respect to the specific humidity shows good correlation with the high specific humidity instead of the precipitation.

Overall, the paper is well written. The science of the paper is significant and it advances our understanding of the liquid/ice water content on the GPS RO measurements, which has puzzled the RO community for quite some time. The discovery of positive N-bias (i.e., negative temperature bias, and/or positive moisture bias) likely attributed to the global analyses/reanalysis in the high specific humidity condition will be worth further investigation. Note some more details related to the simulation of liquid/ice water content are needed in the manuscript, and some text and a few figures can be improved.

I would recommend publication of the paper after "minor revision". The comments are listed below:

Major comments:

1. Section 4 missed some details of the ray-tracing simulation.
   a. The typical range of the liquid and ice water content (along the ray path) derived from TRMM radar reflection measurement (e.g., at different altitude in Fig. 1) should be shown or at least discussed in the manuscript.
   b. The size distribution of the particles, N(D) was not shown.

c. How typical are those precipitation cases observed by TRMM? For example, the size, rain rate, etc, should be discussed.

We have included some typical values about the ray tracing simulation. We provide an order of magnitude of the length of the rays below 15 km and the typical water content paths in the end of Sec 4, and we provide some typical numbers of the 65 TRMM cases in Sec 4.1 for representativeness.

2. The N-bias study (e.g., Fig. 6) focused on the high specific humidity based on COSMIC RO wet retrieval. Could the authors use the analyses/reanalysis specific humidity instead, and see whether the N-bias pattern will remain the same? Or is there any references that compare the RO specific humidity with the global analyses/reanalyses, which confirm the consistency?

One direct consequence of the positive difference between the RO and the analyses refractivity is that the specific humidity difference between RO and analyses is also positive. When the refractivity difference is negative, the specific humidity difference is also negative. The relationship between refractivity difference and specific humidity difference between RO and analyses has been checked, and the correlation is high.

Since the positive bias is larger as q_RO increases (e.g. Fig 5), most points in high regions of q_RO will correspond to points not so high in the q_An (analyses). Most points in the lower values of q_RO would correspond to more similar values of q_An, because the overall bias is smaller. Therefore, the values that contribute the most to the positive bias will be diluted with much more values with less N-difference, reducing the overall bias as a function of q_An. We have checked this relationship, and we observe how the relationship of the bias with q_An is less obvious, specially in the no-rain cases. For the rain cases, the fact that there is an overpopulation in the high region of q_RO (due to the contribution of the tropical precipitation) with respect to the medium values of q makes the refractivity bias as a function of q_An less different than when it is plotted as a function q_RO, with the most obvious difference being that the maximum refractivity difference is reached at lower q_An values.

In consequence, when we repeat Figure 6 using the q_An values the no rain and high specific humidity becomes less populated, and the bias is reduced being significant only for the ERA case. On the other hand, the overall bias in the rain and low specific humidity becomes stronger. The cases with no rain and low specific humidity and rain with high specific humidity keep the main characteristics shown in the q_RO case.

We do not include this discussion in the paper, since we understand that can be explained as a direct consequence of the fact that positive refractivity bias is linked to positive specific humidity differences, which is stated in the discussion. However, we include a citation to Vergados, et al. 2015, which shows that ERA-interim is systematically drier than RO in the tropics in agreement with our reasoning.

3. Figure 3:
   a. Add latitude/longitude labels
   b. Add the title for each row with the precipitation rate "<R>"
   c. Maybe put side labels for "ERA-I, ECMWF_An, GFS"

Corrected

4. Figure 4:
   a. Why the impact height goes all the way down to 0 km, which will likely be ~2km below the earth surface. Should that be "geometric height" instead, especially for the refractivity error plot? Please verify and make sure it is consistent with the manuscript description in L3 in Section 4.1.

It was a mistake. Now the bending angle is shown as a function of the impact height, and refractivity as a function of the geometric height.

5. Figure 6:
   a. Add latitude/longitude labels
   b. Add labels after each column title: "no rain low q (<0.5 g/kg)" etc.
   c. Maybe put side labels for "ERA-I, ECMWF_An, GFS"

Corrected

Technical comments:

The line numbers were messy and not consistent. The following is the best I can do to point to the text in the manuscript.

"liquid and solid water content" -> liquid and ice water content
Corrected

Page-1 – Sec. 1 (right column)
L20: remove "of the"
Removed

Page-3 – Sec.2.1
L01: change to "the global surface precipitation every 30 minutes with a horizontal resolution of 0.1° latitude x 0.1° longitude.
Changed

L10: "compared against" -> "collocated with" ;
Changed

L11: remove "looking for coincident … resulted from such comparison." -> "A total of 16,881 COSMIC RO soundings are identified to be within the swath of the TRMM precipitation measurements (250 km), and within +/- 15 minutes."
Corrected

It's confusing whether the collocation threshold "250 km" (i.e., swath size of TRMM) or "100km" as seen in L19 (Section 2.1)? Please clarify.

There is a difference here between when an event is collocated, i.e. within TRMM swath, and when there is or there is not precipitation in the surroundings of an event, i.e. using information from the closest pixels (within 100 km).

Page-3 – Sec.2.2
L03: spatial … between +/-60 deg and every 30 min -> every 30 minute with spatial coverage between 60°S and 60°N
Changed

L12: remove "(IMERG data is … 30 min)", which is redundant.
Removed

Page-3 – Sec.3
L10: for the two analyses and one reanalysis
Corrected

L18: The hexagon with a diameter of ~30 deg is used in Figure 3. What is the sampling looks like? What the minimum/maximum and average sampling number of the collocation within the hexagon?
If the sampling plot will not be shown, it needs to be mentioned/discussed in the text, to justify the choice of "30 deg".
I would expect it is primarily restricted by the sampling, but could it be possible to reduce the size of hexagon and show better spatial pattern?

It has been added to the text. We have increased the total number of observations including also the 2015 collocated observations, and now the hexagons have a diameter of 20 deg. We have included in the text the minimum and maximum number of observations per bin, and we believe that now it is more clear. With the new diameter of the hexagons, the spatial patterns are clearly seen and can be easily associated to regions known to have heavy precipitation.

L22: Revise sentence: "This figure shows the global distribution of the positive refractivity bias under heavy precipitation. The regional difference of the N-bias is evident and the difference among analyses and reanalysis is also shown."
Revised

Page-4
Figure 2 caption: The compared profiles are …" -> The RO profiles are classified into no-rain (solid) and heavy rain (dashed) based on the collocated GPM IMERG precipitation measurements.
Corrected

The time range should be mentioned in the caption.
Corrected

Page-4 – Sec.3
L10: "The results have shown no …" -> "The results (not shown) reveal no significant changes to the positive N-bias, and confirms that the RO observation geometry is not a contributing factor to the positive bias.
Changed

Page-4 – Sec.4
L21: For the case in this study
Corrected

L22: sum of the effects of all the raindrops
Corrected

L33: Reference needed for "Raindrops fall following gravity and are flattened … oblateshape".
Corrected

L04: What is the subscript 1, 2 refer to? Simply corresponding to "h, and v"? Need to be explained.
Corrected

L06: that is complex -> that is a complex number
Corrected

Page-5 – Sec.4
L35: The REFERENCE is needed for "The N(D) is obtained … TRMM products…"
The reference has been added

L36: provide -> retrieve
Corrected

Page-6 – Sec.4 – Sec. 4.1
L20: It is a bit odd to see the sentence here. "In the left panel of Figure 4, …induced phase delay." It should be moved and integrated into the discussion of Figure 4 in Section 4.1, possibly after L04.
Corrected

Figure 4: I would suggest to use (a, b, c) to identify the three panels, which make it easier to discuss in the manuscript.
We have changed the figure to have a, b, and c labels on top of each panel.

Figure 4 deserves more discussion on each panel, especially the Fig. 4a (SNR plot). For example, what is the precipitation rate of the selected case, where, when, and how big is the precipitation feature? How typical is that compare to the other 62 cases?
A better description of figure 4 is included in Sec. 4.1.

Could the author(s) also add the excess phase delay without rain to be compared to the result (blue shaded) under heavy precipitation?

We have provided the order of magnitude of the total excess phase to be compared with the precipitation induced phase delay.

L08: effect into RO -> effect on RO
Corrected

L432: "The effect that liquid and solid water content has into the RO … is to induce small errors such as those .. Figure 4." -> "the effect of liquid/ice water content on RO refractivity retrieval results in small errors (??%, need numbers), which does not introduce obvious biases in both bending angle and refractivity (Figure 4b, c).
Corrected and errors provided

L435: "Thus, the fact that …scenarios." -> "Thus the scattering terms in Equation 1 should not be the cause for the positive N-bias in the presence of the heavy precipitation."
Corrected

Page-6 – Sec.5
L439 – L457: The two paragraph were not well written and require some revision to improve the points.
Note the author(s) like to use the first person in the manuscript.
These two paragraphs have been rewritten.

L449: "We have done it for the three analyses/reanalyses" -> the two analyses and one reanalysis.
Corrected
L458: "This classification allows us to further .. scenarios:" -> "We further classify the collocated COSMIC RO profiles into four different categories:"
Corrected

L462: the criteria -> the threshold
Corrected

Page-7 – Sec.5
L474: I assume that using coarse resolution of "45 deg" instead of "30 deg" was due to the limited sampling? Please offer the description of the map of sampling after separating into four categories. Will the possibly low and un-even sampling affect the results?

Now the size of the bins is reduced due to the increment of the observations after including year 2015. We have included in the text the typical number of observations per bin at each category. Since the overall values for the bias and its global pattern have not changed significantly after incrementing the number observations, we are confident that the sampling is not affecting the results in a significant way.

L502: "the bias is larger in  of ERA-interim …but smaller in  of ECMWF … analysis, showing the different performance of model in characterizing precipitation.
Corrected

Last paragraph of Sec 5:
Not sure that is a correct interpretation. The physical model used in ERA-I reanalysis and ECMWF analysis should be pretty much the same. The major difference is the spatial resolution as well as the data assimilated. I would argue the resolution might be the major reason behind the difference. Please discuss and justify your reasoning.

We agree with the reviewer that spatial resolution definitely contributes to the differences among analyses. We mention that at the end of Sec 5 and we also add it to the discussion. In fact, here we are not addressing nor we want to reach any conclusion about why analyses and re-analyses perform the way they do. In this study we mainly want to point out that the refractivity bias between RO and analyses exist and, as opposed to previous studies conclusions, it might be due to the analyses and re-analysis information, which can be worth of future investigations.

[revised manuscript text omitted]
  every 30 minutes with a horizontal resolution of 0.1° latitude × 0.1°  longitude.

For this study, the precipitation information comes solely from the TRMM and GPM retrieved products, and no precipitation information is used nor assumed from the analyses and re-analyses. Therefore, the analyses and re-analysis products might or might not be associated to different precipitation conditions when their products are generated. However, it is not the aim of this work to assess the ability of analyses to reproduce precipitation, but to evaluate and compare the RO products with their provided thermodynamic fields for a given location and time in the actual presence or absence of precipitation. Nevertheless, we consider that this could lead to minor effects solely, since the water vapor field is spatio-temporally smoother than the cloud water content field.

**2.1 Collocations with the TRMM PR**

The COSMIC/FORMOSAT-3 RO products between 2006 and 2015 were  collocated with TRMM orbital products . A total of 16,881 COSMIC RO soundings are identified to be within the swath of the TRMM precipitation measurements (250 km), and within  time (both observations within ± 15 minutes. After 2013, the number of COSMIC/FORMOSAT-3 RO observations dropped significantly. However, the quality and distribution of the observations was not affected.  These events were then classified depending on the presence or not of precipitation and its intensity. Henceforth, each event is linked to the number of pixels of the TRMM radar with a reflectivity (Z) larger than 30 dBZ, used as a proxy for heavy precipitation events, in the surroundings (100 km) of the occultation location within the radar swath.

For each event with evidences of precipitation in its surroundings, the approximated RO ray trajectories have been simulated using ray-tracing techniques and geo-located together with the radar retrieved 3 dimensional reflectivity observations. Therefore, it  is possible to interpolate the precipitation information into the set of RO ray trajectories. An example of such interpolation is shown in Figure 1. We can therefore estimate the amount of precipitation crossed by each of the rays, estimate its effect, and compare it with the actual RO observables such as the excess phase (or the Doppler shift), the Signal to Noise Ratio (SNR) or the atmospheric vertical retrievals. We use this information to assess the impact of precipitation into the RO signal propagation and its retrievals, as described in section 4.

It is worth mentioning that in this study we focus on the effect of rain and hydrometeors large enough to exhibit

[Figure]

**Figure 1.** Three dimensional collocation of a RO event with a TRMM PR precipitation measurement. It corresponds to the coincidence between the C004.2006.329.22.20.G19 RO event and the 2B31.20061125.51450 TRMM PR product. Here the precipitation structure is shown in a 3 dimensional grid, along with the set of RO ray trajectories (in gray). Black stars indicate the tangent point of the rays. Only a few rays are shown for illustration purposes. The orange dashed lines indicate the edges of the TRMM PR swath. The interpolated precipitation information (rain rate) into the RO plane is shown in the 2-Dimensional projection in the latitude-height plane.

a significant reflectivity signature in the TRMM radar retrievals (working at Ku-band), which in its turn are the ones we expect to have the largest impact on RO retrievals. The scattering effects of smaller particles, specially above the melting layer, have larger uncertainty and must be treated carefully.

**2.2 Collocations with GPM IMERG**

In order to improve the statistics of collocated profiles we have performed a larger scale collocation using the GPM IMERG products (every 30 minute with spatial coverage between ±60°S and 60°N) and all the COSMIC/FORMOSAT-3 RO products of 2015 and 2016. We can greatly expand the number of collocations by considering only the surface precipitation rate. For each of the COSMIC/FORMOSAT-3 RO events, the corresponding IMERG product has been identified, and the precipitation retrieval has been linked to the RO event. This results in  481,252 RO events from which the surface precipitation in its surroundings has been identified, with a time resolution of ±15min. For each event, the mean rain rate, the maximum rain rate and the number of pixels with non-zero rain rate, in a region of 2° × 2°, is stored along with the vertical RO profiles of refractivity, temperature, pressure, water vapour pressure,

[Figure]

**Figure 2.** Fractional difference between the RO observed refractivity and that from (blue) Era interim re-analysis; (red) ECMWF high resolution analysis; and (green) NCEP GFS operational analysis. The  RO profiles are classified  into no-rain  (solid ) and heavy rain  (dashed )  based on the collocated GPM IMERG  precipitation measurements (see subsection 2.2). These data correspond to 2016.

and the corresponding collocated weather analyses and re-analyses products.

**3   Refractivity bias**

A clear positive refractivity bias is observed between $\sim 3$ and $\sim 8$ km of altitude when precipitation is present in the occultation position, with respect to the refractivity from weather analyses and re-analyses. In Figure 2 the bias is shown, for the comparison between the GPM IMERG collocated RO products and the three different analyses and re-analyses introduced in section 2. In this case, the data are separated according to the amount of rain in the surroundings: events with no rain (no-rain profiles) and events where $\langle R \rangle > 10$ (mm/h) in the $2° \times 2°$ surrounding area.

While the bias is clearly seen for the  for the two analyses and one reanalysis used in the comparison, their performance within heavy precipitation is also different. When precipitation is not present close by the RO sounding, the RO refractivity and that of analyses and re-analyses agree (i.e. no significant bias), as well as among themselves.

In Figure 3 we show the regional dependence of the bias, at a height of 6 km. Through the paper we focus on the height range around 6 km because it is where the bias is maximum, as seen in Figure 2. Here, the globe is divided in hexagons with a diameter of approximately  20 degrees, and the

events are separated according to their $\langle R \rangle$: $\langle R \rangle = 0$ mm/h;  $0 < \langle R \rangle < 2.5$ mm/h; and  $\langle R \rangle > 2.5$ mm/h. This separation is shown at each different column, while the rows separate the analyses or re-analyses used in the comparison. The size of the hexagons is chosen so that all of them contain a significant number of observations and spatial patterns are clearly seen. Only those bins with a minimum of 25 observations inside them are shown, and the typical range of observations inside the bins is between 1,000 and 7,000 observations per bin for the no-rain scenarios, between 200 and 1,600 observations per bin in the low rain regime, and between 25 and 150 observations per bin in the heavy rain regime. This figure shows how the positive bias is present globally under heavy precipitation, although is larger in certain regions, and it depends on the analysis in use. Common features for all three re-analyses are, for example, the positive bias under heavy precipitation that is present in the West Pacific warm pool, the eastern part of the pacific,  Indian ocean, the equatorial part of the Atlantic, and over South America and central Africa. These regions are associated to extreme precipitation features (Liu and Zipser, 2015), either to large extension precipitation events or to precipitation systems with a high deep convective cores.

Besides the positive bias in the region above an altitude of 4 km, a negative bias is also clearly observed below 3 km, both for the rainy and no-rain events. This bias is not assessed here, since it has already been discussed previously in other studies (e.g. Ao et al., 2003; Sokolovskiy, 2003; Xie et al., 2006, 2012; Wang et al., 2017). Similarly, other potential sources of bias have been checked, for example, the angle of incidence of the occultation ray to the receiver, with respect to the transmitter position. The larger the angle, the larger the tangent point drift. This implies that the theoretical spherically symmetric atmosphere could depart from a realistic approximation and induce errors in the retrievals (Foelsche et al., 2011). Also, large incident angles correspond to low SNRs, which could be introducing positive biases (Sokolovskiy et al., 2010). Therefore, the positive bias has been checked grouping the occultation events according to its azimuth angle, in addition to rain variables. The results (not shown) reveal no significant changes  to the positive N-bias, and confirms that the RO observation geometry is not a contributing factor to the positive bias.

**4   Precipitation induced delay**

Once other observational known issues are discarded as plausible sources of the bias, the influence of the scattering term in Equation 1 is assessed. In order to further investigate its importance, we have simulated the contribution of the liquid and solid water directly into the excess phase. This is accomplished using 3-Dimensional collocations between the

[Figure]

**Figure 3.** Regional averaged fractional difference between the RO observed refractivity and that from (top row) Era interim re-analysis; (middle row) ECMWF high resolution analysis; and (bottom row) NCEP GFS operational analysis; for a height of 6 km. The compared profiles are classified between no-rain profiles (left column; $\langle R \rangle = 0$ mm/h), low and moderate precipitation (middle column;  $0 < \langle R \rangle < 2.5$ mm/h) and heavy rain profiles (right column;  $\langle R \rangle > 2.5$ mm/h). The grid corresponds to hexagons with a diameter of about  20 deg. Only those with a minimum of 25 observations inside them are shown.

COSMIC/FORMOSAT-3 RO observations and the TRMM PR measurements, which have allowed us to perform realistic simulations of the effects of precipitation in actual RO observables (see Figure 1). This represents a novel approach to the assessment of the positive refractivity bias with respect to previous studies.

The contribution from precipitation on the phase delay of the signal is due to the scattering of the propagating wave by non-spherical raindrops. The delay induced by raindrops (or frozen hydrometeors) with respect to that of free space can be linked to the scattering term of refractivity in Equation 1. For the case in this study, the coherent propagation of plane waves is described as the sum of the effects  of all the raindrops in a unit volume with various sizes. Formally, the scattered field can be expressed as:

$$E^s = T E^i \tag{2}$$

where $E^i$ is the incident field, $E^s$ is the scattered field, and $T$ is the "transmission matrix" describing the characteristics of the rain medium (Oguchi, 1983). The propagation through rain can be considered as a propagation through an effective medium with two characteristic axes, characterized by the two eigenvalues of $T$, $\lambda_1$ and $\lambda_2$:

$$T = \begin{bmatrix} e^{\lambda_1 l} & 0 \\ 0 & e^{\lambda_2 l} \end{bmatrix} \tag{3}$$

where $l$ is the propagated distance.

Raindrops fall following gravity and are flattened due to the air drag, becoming approximately oblate-shaped (e.g. Pruppacher and Beard, 1970; Beard and Chuang, 1987). Here we do not take into account the canting angle effect (raindrops being tilted by wind), for simplicity and because in this situation its effect is secondary. Therefore, $\lambda_{1,2} = -ik_{\mathrm{eff}}^{h,v}$, where the $k_{\mathrm{eff}}$ is the effective propagation constant of the medium, that is complex  number, and 1 and 2 indicate the characteristic axes of the medium. For the case in this study, the two characteristic axes correspond to $h$ and $v$ (horizontal and vertical).

The effective propagation constant can be expressed as (e.g. Bringi and Chandrasekar, 2001):

$$k_{\mathrm{eff}} = k_0 + \frac{2\pi n_{\mathrm{p}}}{k_0} e_i \boldsymbol{f}(\hat{i}, \hat{i}) \tag{4}$$

where $k_0$ is the propagation constant in the homogeneous atmosphere, $n_{\mathrm{p}}$ is the number of particles per unit volume, $e_i$ indicates the unit polarization vector for the linear states, and $\boldsymbol{f}(\hat{i}, \hat{i})$ is the scattering amplitude vector in the forward scattering configuration. The real part of the effective propagation constant induces a phase shift, while the imaginary part induces an attenuation. At L-band, the attenuation due to the scattering by rain can be neglected. The expression of $k_{\mathrm{eff}}$ is defined for a number of identical particles, but can be generalized to a size distribution of particles defined by $N(D)$. Also, the $\boldsymbol{f}(\hat{i}, \hat{i})$ can be expressed as the Scattering amplitude matrix, S, using the Jones notation (Jones, 1941). The scattering amplitude matrix (2 × 2) relates the scattered field components to the incident field components in the far field approximation. For a right hand circularly polarized (RHCP) propagating field, as it correspond to GNSS transmitted signals, a mean effective propagation constant can be defined by:

$$k_{\text{eff}}^{\text{mean}} = \left( \frac{k_{\text{eff}}^h + k_{\text{eff}}^v}{2} \right), \tag{5}$$

hence, the specific phase shift induced only by the raindrops to a circularly polarized incident wave is:

$$\Delta\Phi^{\text{rain}} = \left( \frac{\lambda}{2\pi} \right) \frac{2\pi}{k_0} \int \Re \left\{ \frac{S_{hh}(D) + S_{vv}(D)}{2} \right\} N(D) dD \tag{6}$$

in units of $\text{mm} \cdot \text{km}^{-1}$, where $\lambda$ is the wavelength (mm), $S_{hh,vv}$ are the co-polar components of the forward scattering amplitude matrix in a linear base of polarization, $N(D)$ is the particle size distribution ($\text{mm}^{-1}\text{m}^{-3}$), and $D$ is the diameter of the particles (mm). The forward scattering amplitude matrix is computed for each scatterer, and depends on the scatterer's size, composition, orientation, and shape (see Bringi and Chandrasekar (2001) for a detailed explanation). For this study, the T-matrix code is used in order to compute $S$ for raindrops of all sizes between 0.1 and 8 mm of diameter (Mishchenko et al., 1996). For the particle shapes, the Beard and Chuang (1987) model is used, which relates the diameter of the each particle with the relationship between its two characteristic dimensions (i.e. its axis ratio). The complex permittivity for liquid water is obtained from Liebe et al. (1991). The $N(D)$ is obtained at each point from the TRMM products ,using the same one used to provide retrieve rain rate from the TRMM PR reflectivity measurements, which is usually approximated with a gamma model (e.g. Kozu et al., 2009).

Using the three dimensional collocations we can therefore compute the phase delay that is solely due to precipitation, in the following way:

- For each collocated event, we have the precipitation information interpolated into the set of RO ray trajectories. The precipitation information (for example, rain rate, water content, etc.), directly or indirectly, is used to infer the $N(D)$ at each point of these trajectories.

- With the $N(D)$, we can compute the specific dΦ^rain $\Delta\Phi^{\text{rain}}$ along each ray using Equation 6, and integrate this quantity along each ray path:

$$\Phi^{\text{rain}} = \int_L \Delta\Phi^{\text{rain}}(l) dl \tag{7}$$

in units of mm, where $L$ is the ray-path length in km.

For each occultation event that has been 3-d collocated with the TRMM PR, we can have the approximate vertical profiles of precipitation induced delay along with all the currently provided information, such as the total excess phase delay, the strength of the signal, and the retrieved vertical thermodynamic products. In the left panel of we show an example of an occultation actual SNR together with the precipitation induced phase delay. To give the reader a context, the ray-paths length below 15 km can be of the order of hundreds of kilometers, and therefore the amount of liquid water content that is crossed is significant. In big precipitating systems, the total integrated liquid water content along the ray-paths can exceed $50 \, \text{kg} \cdot \text{m}^{-2}$.

**4.1 Precipitation induced phase delay impact**

In this section we want to assess the impact that the precipitation induced phase delay has on RO retrievals. To do so we have designed a study that consists in retrieving the bending angle (Phase Matching method (Jensen et al., 2003)) and refractivity (inverse Abel transform (Fjeldbo et al., 1971)) profiles from the total excess phase delay to compare it with the retrieval results when the precipitation induced delays are removed from the original total excess phase. Therefore, the precipitation induced delays obtained in the previous section are removed from the actually observed phase delays, obtaining two profiles called the rain (original) and the rain-free (where the precipitation induced delay has been removed).

The bending angle and refractivity retrieval were attempted on both rain and rain-free excess phases from a total of 65 cases collocated with heavy precipitation events. The bending angle profiles calculated by Phase Matching were smoothed with 200m windows and compared in the same impact height (corresponding impact parameter minus the collocated radius of Earth).

An example of one of the 65 collocated cases is shown in Figure 4. In panel (a) of Figure 4 we show an example of the actual occultation SNR (black) together with the precipitation induced phase delay (blue shaded), as a function of time with respect to the start of the occultation. Note that the maximum precipitation induced phase delay is of the order of hundreds of millimeters, while the total excess phase at the lower layers of the atmosphere is of the order of kilometers. This case corresponds to a precipitating cell in the Indian ocean (11°N and 72°E), with an approximate extension of 10,000 km² and rain rate exceeding 20 mm/h. The combination of extension and intensity makes this case an interesting one, inducing an excess phase larger than 110 mm. The cases selected for this work are those with the largest precipitation induced excess phase, and are all around 100 mm. They are representative of the variety of collocated cases, combining different intensities and extensions.

In panels (b) and (c) of Figure 4 we show the mean (black line) and standard deviation (orange shade) of the difference between the retrieval using the actual measurements and

[Figure]

**Figure 4.** (left) Actual SNR (black) corresponding to the RO event C001.2008.345.00.43.G03 (UCAR id), along with the simulated precipitation induced phase delay (blue) as a function of time; (right) Fractional bending angle and refractivity differences between the outputs from the retrieval using the rain-affected profiles and the rain-removed ones, as a function of the impact height (bending angle) and of the geometric height (refractivity). Black lines represent the mean of the 65 cases, while orange shade is the standard deviation.

those after removing the precipitation induced excess phase, both for bending angle (panel b) as a function of the impact height and refractivity (panel c) as a function of geometric height. Because of the integration nature of inverse Abel transform, the standard deviation (orange shade in  panels (b) and (c) of Figure 4) in the retrieved refractivity is much smoother than the one in bending angle profiles. If precipitation had a systematic effect  on RO phase delays, a positive bias would be observed in the rain-affected bending angle and refractivity when compared with the rain free bending and refractivity for the same case. However, this effect is absent in the right panels of Figure 4.

The results of nonexistent mean positive bias shown in the right panels of Figure 4 suggest that the positive bias found in the retrieved refractivity compared to the weather analyses and re-analyses is not due to the neglect of the scattering term in the refractivity. Furthermore, it can be seen how on average, the impact of taking / not taking into account the precipitation induced delays when assessing the retrieval increases the variability, implying that the changes of removing precipitation contribution from the signal propagation can be both positive and negative, rather than only negative. Since the bending angle and refractivity retrieval process depends mostly on the vertical gradient of the excess phase, the precipitation induced excess Doppler, which can be positive or negative, will on average lead to unbiased retrieval results. This extra excess Doppler can be seen as the result of local horizontal inhomogeneity in the refractivity field.

Differently from temperature and pressure, the liquid and ice water content is localized in a small region (compared to the ray travel distance), and might not be contributing along the whole propagation ray-path of an occultation. Furthermore, the region where liquid and ice water is present might be far from the tangent point. Yet, the refractivity retrieved from a RO observation is located around the RO tangent point, and considered to have an horizontal resolution of about 200 km (e.g. Kursinski et al., 1997). Even though the RO observations are integral quantities, most of the contribution from dry and wet air in the bending angle comes from near the tangent point.

In addition, the RO retrievals rely on the spherical symmetric atmosphere approximation. While it has been proven to work properly for the standard RO thermodynamic products, liquid and  ice water content contributions to the excess phase cannot be well captured under such assumption. In consequence, the effect  of liquid/ice water content on RO refractivity retrieval results in small errors (e.g. less than 1% in refractivity standard deviation), which does not introduce obvious biases in both bending angle and refractivity Figure 4. Thus  the scattering terms in Equation 1  should not be the cause for the positive  N-bias in the presence of heavy precipitation.

**5 Specific humidity as a source of refractivity bias**

 In the previous section we have shown that the scattering term in Equation 1  should not be the main cause for the refractivity bias. In this section we test another hypothesis: the bias comes from the problems of large scale analyses and re-analyses  in representing the thermodynamics of precipitation scenarios, specially under high specific humidity conditions.

 We have used the data described in subsection 2.2  to assess the refractivity bias as a function of the RO retrieved specific humidity    , for precipitating and non-precipitating scenarios. In Figure 5 we show the results for the two analyses (ECMWF high res-olution    and GFS operational analyses) and one re-analysis (ERA-Interim). We can see how the fractional refractivity difference increase with specific hu-midity  regardless of precipitation. Therefore, the re-fractivity bias is  more correlated with increasing specific humidity than with precipitation itself. However, high specific humidity condi-tions are  associated to precipitation.

 We further classify the collocated COSMIC RO profiles into four different categories: no rain with low specific humidity conditions; no rain with high spe-cific humidity; rain with low specific humidity; and rain with high specific humidity. In this case, the  threshold for low and high specific humidity is that the RO retrieved $q$ is lower than 0.5 g/kg and higher than 2.7 g/kg, respectively, in the cases with no rain, and that the RO retrieved $q$ is lower than  0.5 g/kg and higher than 2.7 g/kg, respectively, in the cases with rain. The $q$ and the fractional refractivity dif-ference are evaluated at a height of 6.5 km. These thresholds are based on the lower and higher 20th and 80th percentiles of data with no rain and rain. For these four classifications, the regional fractional refractivity differences are shown in Figure 6, for the comparison with ERA-interim re-analysis, ECMWF high resolution analysis and the GFS analysis. Here the globe is divided in hexagons of a diameter of 30 deg, and only those with a minimum of 15 observations inside are shown. The typical range of observations per bin is between 800 and 10,000 for the no rain with low specific humidity scenarios, between 15 and 600 for the no rain and high specific humidity regimes, between 15 and 80 for the rain with low specific humidity, and between 15 and 250 for the rain and high specific humidity regimes.

The results in Figure 6 confirm the results anticipated in Figure 5, i.e. the fractional refractivity bias can be linked to high specific humidity conditions rather than to precipitation itself. From the regional dependence of the fractional refrac-tivity bias some other conclusions can be extracted. The first one is that when there is no rain and the specific humidity is low, the fractional refractivity difference is very small re-gardless of location and the analyses in use.

The second conclusion one can extract from Figure 6 is that when specific humidity is high, the fractional refractiv-ity difference is positive and reaches large values ($> 1\%$), for all the analyses in use and regardless of the presence of pre-

[Figure]

**Figure 5.** Fractional refractivity difference between the observa-tions and analyses as a function of the observed specific humidity at a height of 6 km. The left column corresponds to the no-rain cases and the right column corresponds to the rain cases. The top row shows the results for the comparison of observations and ERA-interim, the middle row show the results for ECMWF high resolu-tion analysis and the bottom row shows the results for the GFS.

cipitation. In particular, high specific humidity observations are concentrated in the tropics, so the largest positive refrac-tivity bias are in this region, in agreement with Figure 3.

The third conclusion is that precipitation under low spe-cific humidity conditions is rarely observed in the tropics. Under these conditions, the fractional refractivity difference has a more complicated behavior   and no clear positive bias is observed, but a variability depending on the location of the observations.

Finally,  in addition to the positive frac-tional refractivity difference bias  linked to high spe-cific humidity conditions, we show that it is also depen-dent on the analysis in use. For example, the bias is larger in  ERA-Interim re-analysis

[Figure]

**Figure 6.** Regional averaged differences (colorscale) between the observed and the analysis refractivity. The first row corresponds to ERA interim, the middle row corresponds to ECMWF high resolution and the bottom one to the GFS. The two left rows corresponds to the free of rain data, where in the first row the observed specific humidity at a height of 6.5 km is lower than 0.5 g/kg, and in the second row it is larger than 2.7 g/kg. The two right columns represent rain affected data, where in the third column the observed specific humidity at a height of 6.5 km is lower than  0.5 g/kg, and in the last row it is larger than 2.7 g/kg. The grid here corresponds to hexagons with a diameter of  30 deg. Only those with a minimum of 15 observations inside them are shown.

 than in ECMWF high resolution analysis and GFS operational analysis, showing the different performance of  the analyses and re-analysis, with a smaller bias for the higher resolution analyses. On the other hand, for no rain and low specific humidity, the performance of the different analyses and re-analysis is similar.  Overall, the fact that the  bias is positive is an indication that models tend to be biased dry . This is in agreement with Hersbach et al. (2015),  who noticed a dry bias in ERA-Interim which was attributed to a problem in assimilating microwave  radiances affected by rain.

**6 Summary and discussion**

[revised manuscript text omitted]

---

## Author Comment (AC2) · 24 Jul 2018

Reviewer #2:

The authors would like to thank the reviewers for spending time and effort reviewing this manuscript. Their comments are deeply appreciated and have resulted in an improved version of the manuscript. Below we address point by point all the comments and recommendations of the reviewers, and at the end of the document there is the new version of the paper with the changes highlighted in blue.

This manuscript investigated the underlying physical cause of a systematic high-bias in retrieved GNSS radio occultation refractivity in the middle troposphere under heavy precipitating situations. Previously people had found similar bias signals but attributed the bias to the scattering caused by raindrop/frozen hydrometers. In this study, the authors interpolated the collocated 3D TRMM-precipitation radar retrieved rain rate to the GPS RO plane and found the difference between simulated SNR and observed ones are not systematically biased. Rather, the authors found some positive relationships between the percentage bias (%) and the collocated specific humidity from three different re-analysis/analyses products. Therefore, the conclusion they made from this work is that the systematic high-bias under heavy precipitation scenes are caused by the corresponded increase in specific humidity.

This idea is novel. The comparisons to multiple observations and reanalysis/analyses products make the conclusion rather solid. The writing is clear and concise. I think this manuscript deserves the final publication in ACP. However, there are some logic caveats I'd like to point out that are either not considered clearly enough when carrying out the data analysis, or not described clear enough to make the readers not confused. There are a few minor glitches that may improve after the revision.

First, with respect to the broad picture of the logic flow:
(1) the heavy precipitation scenes are defined by TRMM-PR or IMERG "observations" (by saying observations, I mean retrieved products), but not the precipitation scenes in the reanalysis/analyses products. However, the collocated specific humidity profiles are identified from the re-analysis/analyses. Therefore, when you separate the specific humidity value according to no-rain, light-rain and heavy-rain, it's not necessarily the specific humidity environment for no-rain/light-rain/heavy-rain in the reanalysis/analyses products. I believe this effect is minor as the water vapor field is rather smooth and not as intermittent as the cloud water content field. But I think you need to be clear about this logic difference in the manuscript.

It is true that for this study we do not take into account the precipitation information from the analyses or re-analyses. For this work we only want to evaluate the differences between RO and the analyses (which are generally used as first guesses for RO products) in reproducing the state of the atmosphere at a given time and location for a given precipitation conditions. We have included this information in the manuscript, so now it is clear.

(2) secondly, regarding the collocation and co-incident measurements between GPS-RO and TRMM-PR and IMERG: as precipitation is so transient, especially for heavy precipitation, +/- 15 minutes criterion for co-incident measurements might be too loose. The geo-collocation criterion for TRMM-PR and IMERG was not specified clearly in the context: Do you consider the footprint

effect? How do you align the GPS-RO limb-sounding with TRMM-PR type of nadir-viewing instrument?

We agree with the reviewer that the time difference between observations has to be treated carefully. However, in this study we are interested in the realistic magnitude of the precipitation induced excess phase. For the 3 dimensional analysis we use 65 cases where the induced excess phase is substantial, even though it is possible that it is not the actual induced excess phase due to the time difference (statistically, we should be underestimating as well as overestimating). Still, removing a realistic contribution of precipitation induced differential phase delay should lead to a bias between the two different retrievals (rain contributed vs rain removed), if the scattering with precipitation was the cause of the positive bias. This is also because the water vapor field is smooth, and therefore the retrieved thermodynamic variables should not change too much within the time range (15 min).

Regarding the geo-colocation criterion, we have rephrased a few sentences in Sec 2 that we hope make the method more understandable. The basic idea is that Radio Occultations provide a vertical scanning of the atmosphere through a set of limb-sounding rays, which are simulated and geolocated into the radar retrieved reflectivity field. Then, the reflectivity measurements can be interpolated into the rays, whose contribution is integrated along each of the rays.

(3) the simulation of heavy precipitation scenes using the assumed raindrop size and size distribution is more or less questionable to me. As above 5 km melting layer, most of the precipitation-sized particles are frozen hydrometers indeed, and using raindrop assumption throughout the column is not a very good assumption. But once you calculate scattering frozen hydrometers, more free parameters like ice habit, density, etc., will be involved and the uncertainties are huge. I think it is NOT a good idea to COMPLETELY exclude the precipitation-sized particle scattering effect out, first because of the aforementioned bullet#1, and also because simulation of scattering effect for any hydrometers, especially when frozen hydrometers are involved, is far from perfect. Actually, since the largest discrepancy occurs at ~5 km in Fig. 1 for all three re-analyses/analyses datasets, I suspect strongly that the extremely complicated situation in the melting layer (at ~5km in the tropics) would at least have certain effect on that.

We agree with the reviewer that scattering effects cannot be completely excluded, as we explain in the third paragraph of the discussion. We have included a paragraph in Sec 2 mentioning the fact that we are only taking into account those hydrometeors being detected by TRMM, and saying that specially above the melting layer the scattering effects can have larger uncertainties.

The drop size distribution used for this study is based on the three dimensional retrievals of reflectivity from TRMM, therefore it changes with altitude and accounts for melting layer as well as other vertical structures, if any. Specially in the tropics, where deep convective precipitation cells are expected, significant values of reflectivity are obtained above 5 km.

In addition, the excess phase delay induced solely by hydrometeors strongly depends on the scattering amplitude matrix, which in turn depends on the composition of the medium. In the case of ice, the permittivity is one order of magnitude smaller than that of liquid water in the frequency range in use, and therefore we expect its contribution to be smaller.

Finally, the fact that the bias is maximum around 5 km (or in the free troposphere) could also be related with the findings of e.g. Holloway and Neelin 2009, suggesting that the free tropospheric water vapor (i.e. above the boundary layer up to ~200mb) has a main role controlling heavy

precipitation and that this dependence is not well represented in global climate models. A comment on this is also included in the discussion.

(4) Since the authors didn't discuss throughout the manuscript that why they reach different conclusions with previous literatures (e.g., Lin et al., 2010; Yang and Zou, 2012; etc. as mentioned in the manuscript), I'm not sure whether it's because the analysis methodology is different? Data sources are different? Assumptions are different? It worths a paragraph or at least a couple of sentences to discuss the differences in your and previous efforts that eventually lead to the discrepancy in conclusions.

We believe that different conclusions are obtained because we used different approach. The approach is briefly summarized in the introduction (and then extensively treated in Sec 4) when we talk about previous studies, that is, we assess the effect of precipitation directly in the RO observables using the 3D collocations and how these observables lead to the RO retrievals. We have now also emphasized the fact that we are using the different approach in the discussion.

(5) As I mentioned in Comment#3 above, this paper didn't explain the vertical structure of the high-bias shown in Fig. 1. Rather, the last two figures (Fig.5 and 6) and related discussions focus on a single altitude (6 km or 6.5 km) and the reason why this altitude is selected was not specified clearly in the text.

The height range is selected where the bias appears to be maximum, according to Figure 2. Now it is clearly specified in the text. We also believe that the height at which the bias is maximum can have something to do with the free-tropospheric water vapor being not well represented in models, as we discuss at the end of reviewer's comment #3.

Minor points:
Page 8, equation (7): please be consistent with dphi or selta_phi.
Corrected

Figure 3 and Figure 6: Since the collocated sample for heavy precipitation scenes is small, it's important to have the statistical significance level shown on the map. Please consider only color statistically significant grids, or overplot the contoured significance level.

We have increased the number of observations with respect to the previous version by including the 2015 data. Now we only show the bins with a significant number of observations inside (now detailed in the text), and we also detail the range of minimum and maximum observations per bin in each Figure. The patterns remain the same as the previous version, therefore the bias and the patterns are not associated to the number of samples in each bin.

Figure 2: Do you have any speculation why ERA-rain looks worse than ECH-rain?

One main reason could be the spatial resolution. We have included this at the end of Sec 5 and in the discussion.

Also, please include the standard deviation envelope for each rain curve.

We prefer not to include the standard deviation envelopes for the curves. This would result with a rather complicated figure, that could lead the readers to misunderstandings. It would be better to show the standard error, but due to the large number of samples it is small, and we do not include it in the figure.

Figure 4: Can you show a case with negative delay of SNR, together with corresponding TRMM-PR rain rate vertical profile projected on the RO plane for the two cases? I don't get in what situation that SNR delay could be reversed.

Indeed, the SNR cannot be reversed. What can be positive and negative is the difference between the actual and the rain-removed retrieved bending angle and refractivity, and that is what the 0-centered means in the right panels of figure 4 show. This is because the retrievals depend on the vertical gradient rather than the absolute values. We explain this in the 5th paragraph in Sec 4.1. Therefore, when the excess phase delay induced by precipitation decrease as a function of time (or height), there may exist a negative difference in the bending angle or refractivity profiles.

Regarding Fig. 5 and Fig. 3: if your hypothesis is correct, you should see smaller specific humidity for light-precipitation scenes (Fig. 3, middle column). Is that the case from the collocated re-analysis/analysis?

Yes. When we plot an histogram of both RO and analyses specific humidity we observe a shorter tail in the cases of light precipitation than in the cases of heavy precipitation, for both specific humidity datasets. Overall, RO specific humidity tends to reach higher values, consistent with the fact that when the refractivity difference is positive, specific humidity difference between RO and analyses is also positive. The relationship between the refractivity bias and specific humidity difference has been also checked, and the correlation is very high, as it is expected.

Page 11, Line 22: I don't understand this statement. Would you please elaborate?
The last paragraph of Sec 5 has been rewritten.

[revised manuscript text omitted]

5  longitude.

For this study, the precipitation information comes solely from the TRMM and GPM retrieved products, and no precipitation information is used nor assumed from the analyses and re-analyses. Therefore, the analyses and
10 re-analysis products might or might not be associated to different precipitation conditions when their products are generated. However, it is not the aim of this work to assess the ability of analyses to reproduce precipitation, but to evaluate and compare the RO products with their provided
15 thermodynamic fields for a given location and time in the actual presence or absence of precipitation. Nevertheless, we consider that this could lead to minor effects solely, since the water vapor field is spatio-temporally smoother than the cloud water content field.

20 ## 2.1 Collocations with the TRMM PR

The COSMIC/FORMOSAT-3 RO products between 2006 and 2015 were  collocated with TRMM orbital products. A total of 16,881 COSMIC RO soundings
25 are identified to be within the swath of the TRMM precipitation measurements (250 km), and within  time (both observations within ± 15 minutes. After 2013, the number of COSMIC/FORMOSAT-3 RO observations dropped significantly. However, the qual-
30 ity and distribution of the observations was not affected.  These events were then classified depending on the presence or not of precipitation and its intensity. Henceforth, each event is linked to the number of pixels of the TRMM radar
35 with a reflectivity (Z) larger than 30 dBZ, used as a proxy for heavy precipitation events, in the surroundings (100 km) of the occultation location within the radar swath.

For each event with evidences of precipitation in its surroundings, the approximated RO ray trajectories have
40 been simulated using ray-tracing techniques and geo-located together with the radar retrieved 3 dimensional reflectivity observations. Therefore, it  is possible to interpolate the precipitation information into the set of RO ray trajectories. An example of such interpolation is shown in Fig-
45 ure 1. We can therefore estimate the amount of precipitation crossed by each of the rays, estimate its effect, and compare it with the actual RO observables such as the excess phase (or the Doppler shift), the Signal to Noise Ratio (SNR) or the atmospheric vertical retrievals. We use this information
50 to assess the impact of precipitation into the RO signal propagation and its retrievals, as described in section 4.

It is worth mentioning that in this study we focus on the effect of rain and hydrometeors large enough to exhibit

[Figure]

**Figure 1.** Three dimensional collocation of a RO event with a TRMM PR precipitation measurement. It corresponds to the coincidence between the C004.2006.329.22.20.G19 RO event and the 2B31.20061125.51450 TRMM PR product. Here the precipitation structure is shown in a 3 dimensional grid, along with the set of RO ray trajectories (in gray). Black stars indicate the tangent point of the rays. Only a few rays are shown for illustration purposes. The orange dashed lines indicate the edges of the TRMM PR swath. The interpolated precipitation information (rain rate) into the RO plane is shown in the 2-Dimensional projection in the latitude-height plane.

a significant reflectivity signature in the TRMM radar retrievals (working at Ku-band), which in its turn are the
55 ones we expect to have the largest impact on RO retrievals. The scattering effects of smaller particles, specially above the melting layer, have larger uncertainty and must be treated carefully.

**2.2 Collocations with GPM IMERG                                              60**

In order to improve the statistics of collocated profiles we have performed a larger scale collocation using the GPM IMERG products (every 30 minute with spatial coverage between ±60°S and 60°N) and all the COSMIC/FORMOSAT-3 RO products of 2015 and 2016.
65 We can greatly expand the number of collocations by considering only the surface precipitation rate. For each of the COSMIC/FORMOSAT-3 RO events, the corresponding IMERG product has been identified, and the precipitation retrieval has been linked to the RO event. This results in
70  481,252 RO events from which the surface precipitation in its surroundings has been identified, with a time resolution of ±15min. For each event, the mean rain rate, the maximum rain rate and the number of pixels with non-zero rain rate, in a re-
75 gion of 2° × 2°, is stored along with the vertical RO profiles of refractivity, temperature, pressure, water vapour pressure,

[Figure]

**Figure 2.** Fractional difference between the RO observed refractivity and that from (blue) Era interim re-analysis; (red) ECMWF high resolution analysis; and (green) NCEP GFS operational analysis. The  RO profiles are classified  into no-rain  (solid ) and heavy rain  (dashed )  based on the collocated GPM IMERG  precipitation measurements (see subsection 2.2). These data correspond to 2016.

and the corresponding collocated weather analyses and re-analyses products.

**3 Refractivity bias**

A clear positive refractivity bias is observed between $\sim 3$ and $\sim 8$ km of altitude when precipitation is present in the occultation position, with respect to the refractivity from weather analyses and re-analyses. In Figure 2 the bias is shown, for the comparison between the GPM IMERG collocated RO products and the three different analyses and re-analyses introduced in section 2. In this case, the data are separated according to the amount of rain in the surroundings: events with no rain (no-rain profiles) and events where $\langle R \rangle > 10$ (mm/h) in the $2° \times 2°$ surrounding area.

While the bias is clearly seen for the  for the two analyses and one reanalysis used in the comparison, their performance within heavy precipitation is also different. When precipitation is not present close by the RO sounding, the RO refractivity and that of analyses and re-analyses agree (i.e. no significant bias), as well as among themselves.

In Figure 3 we show the regional dependence of the bias, at a height of 6 km. Through the paper we focus on the height range around 6 km because it is where the bias is maximum, as seen in Figure 2. Here, the globe is divided in hexagons with a diameter of approximately  20 degrees, and the

events are separated according to their $\langle R \rangle$: $\langle R \rangle = 0$ mm/h;  $0 < \langle R \rangle < 2.5$ mm/h; and  $\langle R \rangle > 2.5$ mm/h. This separation is shown at each different column, while the rows separate the analyses or re-analyses used in the comparison. The size of the hexagons is chosen so that all of them contain a significant number of observations and spatial patterns are clearly seen. Only those bins with a minimum of 25 observations inside them are shown, and the typical range of observations inside the bins is between 1,000 and 7,000 observations per bin for the no-rain scenarios, between 200 and 1,600 observations per bin in the low rain regime, and between 25 and 150 observations per bin in the heavy rain regime. This figure shows how the positive bias is present globally under heavy precipitation, although is larger in certain regions, and it depends on the analysis in use. Common features for all three re-analyses are, for example, the positive bias under heavy precipitation that is present in the West Pacific warm pool, the eastern part of the pacific,  Indian ocean, the equatorial part of the Atlantic, and over South America and central Africa. These regions are associated to extreme precipitation features (Liu and Zipser, 2015), either to large extension precipitation events or to precipitation systems with a high deep convective cores.

Besides the positive bias in the region above an altitude of 4 km, a negative bias is also clearly observed below 3 km, both for the rainy and no-rain events. This bias is not assessed here, since it has already been discussed previously in other studies (e.g. Ao et al., 2003; Sokolovskiy, 2003; Xie et al., 2006, 2012; Wang et al., 2017). Similarly, other potential sources of bias have been checked, for example, the angle of incidence of the occultation ray to the receiver, with respect to the transmitter position. The larger the angle, the larger the tangent point drift. This implies that the theoretical spherically symmetric atmosphere could depart from a realistic approximation and induce errors in the retrievals (Foelsche et al., 2011). Also, large incident angles correspond to low SNRs, which could be introducing positive biases (Sokolovskiy et al., 2010). Therefore, the positive bias has been checked grouping the occultation events according to its azimuth angle, in addition to rain variables. The results (not shown) reveal no significant changes  to the positive N-bias, and confirms that the RO observation geometry is not a contributing factor to the positive bias.

**4 Precipitation induced delay**

Once other observational known issues are discarded as plausible sources of the bias, the influence of the scattering term in Equation 1 is assessed. In order to further investigate its importance, we have simulated the contribution of the liquid and solid water directly into the excess phase. This is accomplished using 3-Dimensional collocations between the

[revised manuscript text omitted]

- For each collocated event, we have the precipitation information interpolated into the set of RO ray trajectories. The precipitation information (for example, rain rate, water content, etc.), directly or indirectly, is used to infer the $N(D)$ at each point of these trajectories.

- With the $N(D)$, we can compute the specific $d\Phi^{\text{rain}}$ $\Delta\Phi^{\text{rain}}$ along each ray using Equation 6, and integrate this quantity along each ray path:

$$\Phi^{\text{rain}} = \int_{L} \Delta\Phi^{\text{rain}}(l)dl \tag{7}$$

in units of mm, where $L$ is the ray-path length in km.

For each occultation event that has been 3-d collocated with the TRMM PR, we can have the approximate vertical profiles of precipitation induced delay along with all the currently provided information, such as the total excess phase delay, the strength of the signal, and the retrieved vertical thermodynamic products. In the left panel of we show an example of an occultation actual SNR together with the precipitation induced phase delay. To give the reader a context, the ray-paths length below 15 km can be of the order of hundreds of kilometers, and therefore the amount of liquid water content that is crossed is significant. In big precipitating systems, the total integrated liquid water content along the ray-paths can exceed $50 \text{ kg} \cdot \text{m}^{-2}$.

**4.1 Precipitation induced phase delay impact**

In this section we want to assess the impact that the precipitation induced phase delay has on RO retrievals. To do so we have designed a study that consists in retrieving the bending angle (Phase Matching method (Jensen et al., 2003)) and refractivity (inverse Abel transform (Fjeldbo et al., 1971)) profiles from the total excess phase delay to compare it with the retrieval results when the precipitation induced delays are removed from the original total excess phase. Therefore, the precipitation induced delays obtained in the previous section are removed from the actually observed phase delays, obtaining two profiles called the rain (original) and the rain-free (where the precipitation induced delay has been removed).

The bending angle and refractivity retrieval were attempted on both rain and rain-free excess phases from a total of 65 cases collocated with heavy precipitation events. The bending angle profiles calculated by Phase Matching were smoothed with 200m windows and compared in the same impact height (corresponding impact parameter minus the collocated radius of Earth).

An example of one of the 65 collocated cases is shown in Figure 4. In panel (a) of Figure 4 we show an example of the actual occultation SNR (black) together with the precipitation induced phase delay (blue shaded), as a function of time with respect to the start of the occultation. Note that the maximum precipitation induced phase delay is of the order of hundreds of millimeters, while the total excess phase at the lower layers of the atmosphere is of the order of kilometers. This case corresponds to a precipitating cell in the Indian ocean (11°N and 72°E), with an approximate extension of 10,000 km² and rain rate exceeding 20 mm/h. The combination of extension and intensity makes this case an interesting one, inducing an excess phase larger than 110 mm. The cases selected for this work are those with the largest precipitation induced excess phase, and are all around 100 mm. They are representative of the variety of collocated cases, combining different intensities and extensions.

In panels (b) and (c) of Figure 4 we show the mean (black line) and standard deviation (orange shade) of the difference between the retrieval using the actual measurements and

[Figure]

**Figure 4.** (left) Actual SNR (black) corresponding to the RO event C001.2008.345.00.43.G03 (UCAR id), along with the simulated precipitation induced phase delay (blue) as a function of time; (right) Fractional bending angle and refractivity differences between the outputs from the retrieval using the rain-affected profiles and the rain-removed ones, as a function of the impact height (bending angle) and of the geometric height (refractivity). Black lines represent the mean of the 65 cases, while orange shade is the standard deviation.

those after removing the precipitation induced excess phase, both for bending angle (panel b) as a function of the impact height and refractivity (panel c) as a function of geometric height. Because of the integration nature of inverse Abel transform, the standard deviation (orange shade in  panels (b) and (c) of Figure 4) in the retrieved refractivity is much smoother than the one in bending angle profiles. If precipitation had a systematic effect  on RO phase delays, a positive bias would be observed in the rain-affected bending angle and refractivity when compared with the rain free bending and refractivity for the same case. However, this effect is absent in the right panels of Figure 4.

The results of nonexistent mean positive bias shown in the right panels of Figure 4 suggest that the positive bias found in the retrieved refractivity compared to the weather analyses and re-analyses is not due to the neglect of the scattering term in the refractivity. Furthermore, it can be seen how on average, the impact of taking / not taking into account the precipitation induced delays when assessing the retrieval increases the variability, implying that the changes of removing precipitation contribution from the signal propagation can be both positive and negative, rather than only negative. Since the bending angle and refractivity retrieval process depends mostly on the vertical gradient of the excess phase, the precipitation induced excess Doppler, which can be positive or negative, will on average lead to unbiased retrieval results. This extra excess Doppler can be seen as the result of local horizontal inhomogeneity in the refractivity field.

Differently from temperature and pressure, the liquid and ice water content is localized in a small region (compared to the ray travel distance), and might not be contributing along the whole propagation ray-path of an occultation. Furthermore, the region where liquid and ice water is present might be far from the tangent point. Yet, the refractivity retrieved from a RO observation is located around the RO tangent point, and considered to have an horizontal resolution of about 200 km (e.g. Kursinski et al., 1997). Even though the RO observations are integral quantities, most of the contribution from dry and wet air in the bending angle comes from near the tangent point.

In addition, the RO retrievals rely on the spherical symmetric atmosphere approximation. While it has been proven to work properly for the standard RO thermodynamic products, liquid and  ice water content contributions to the excess phase cannot be well captured under such assumption. In consequence, the effect  of liquid/ice water content on RO refractivity retrieval results in small errors (e.g. less than 1% in refractivity standard deviation), which does not introduce obvious biases in both bending angle and refractivity Figure 4. Thus  the scattering terms in Equation 1  should not be the cause for the positive  N-bias in the presence of heavy precipitation.

**5 Specific humidity as a source of refractivity bias**

 In the previous section we have shown that the scattering term in Equation 1  should not be the main cause for the refractivity bias In this section we test another hypothesis: the bias comes from the problems of large scale analyses and re-analyses  in representing the thermodynamics of precipitation scenarios, specially under high specific humidity conditions.

Using We have used the data described in subsection 2.2 , we have studied to assess the refractivity bias as a function of the RO retrieved specific humidity. In turn, these cases are also separated between whether precipitation was present in the surroundings of the observation or not. We have done it for the three analyses /re-analyses: the ERA-interim, the , for precipitating and non-precipitating scenarios. In Figure 5 we show the results for the two analyses (ECMWF high resolution analysis and the GFS , and the results are shown in . Revealing results can be found here: the refractivity fractional difference increases and GFS operational analyses) and one re-analysis (ERA-Interim). We can see how the fractional refractivity difference increase with specific humidity , regardless of precipitation. Hence Therefore, the refractivity bias is linked to high specific humidity rather than more correlated with increasing specific humidity than with precipitation itself. However, high specific humidity conditions are strongly correlated with associated to precipitation.

This classification allows us to further investigate four different scenarios We further classify the collocated COSMIC RO profiles into four different categories: no rain with low specific humidity conditions; no rain with high specific humidity; rain with low specific humidity; and rain with high specific humidity. In this case, the criteria threshold for low and high specific humidity is that the RO retrieved $q$ is lower than 0.5 g/kg and higher than 2.7 g/kg, respectively, in the cases with no rain, and that the RO retrieved $q$ is lower than 1.0 0.5 g/kg and higher than 2.7 g/kg, respectively, in the cases with rain. The $q$ and the fractional refractivity difference are evaluated at a height of 6.5 km. These thresholds are based on the lower and higher 20th and 80th percentiles of data with no rain and rain. For these four classifications, the regional fractional refractivity differences are shown in Figure 6, for the comparison with ERA-interim re-analysis, ECMWF high resolution analysis and the GFS analysis. Here the globe is divided in hexagons of a diameter of 45 deg. 30 deg, and only those with a minimum of 15 observations inside are shown. The typical range of observations per bin is between 800 and 10,000 for the no rain with low specific humidity scenarios, between 15 and 600 for the no rain and high specific humidity regimes, between 15 and 80 for the rain with low specific humidity, and between 15 and 250 for the rain and high specific humidity regimes.

The results in Figure 6 confirm the results anticipated in Figure 5, i.e. the fractional refractivity bias can be linked to high specific humidity conditions rather than to precipitation itself. From the regional dependence of the fractional refractivity bias some other conclusions can be extracted. The first one is that when there is no rain and the specific humidity is low, the fractional refractivity difference is very small regardless of location and the analyses in use.

The second conclusion one can extract from Figure 6 is that when specific humidity is high, the fractional refractivity difference is positive and reaches large values ($> 1\%$), for all the analyses in use and regardless of the presence of pre-

[Figure]

**Figure 5.** Fractional refractivity difference between the observations and analyses as a function of the observed specific humidity at a height of 6 km. The left column corresponds to the no-rain cases and the right column corresponds to the rain cases. The top row shows the results for the comparison of observations and ERA-interim, the middle row show the results for ECMWF high resolution analysis and the bottom row shows the results for the GFS.

cipitation. In particular, high specific humidity observations are concentrated in the tropics, so the largest positive refractivity bias are in this region, in agreement with Figure 3.

The third conclusion is that precipitation under low specific humidity conditions is rarely observed in the tropics. Under these conditions, the fractional refractivity difference has a more complicated behavior , more model dependent than the rest of the cases. In this case, and no clear positive bias is observed, but a variability depending on the location of the observations.

The only situation with a prominent negative bias is observed under this scenario in the west area of Africa.

Finally, even though a in addition to the positive fractional refractivity difference bias can be linked to high specific humidity conditions, we show that it is also dependent on the analysis in use. For example, the bias is larger in the case of ERA-interim ERA-Interim re-analysis , but

[Figure]

**Figure 6.** Regional averaged differences (colorscale) between the observed and the analysis refractivity. The first row corresponds to ERA interim, the middle row corresponds to ECMWF high resolution and the bottom one to the GFS. The two left rows corresponds to the free of rain data, where in the first row the observed specific humidity at a height of 6.5 km is lower than 0.5 g/kg, and in the second row it is larger than 2.7 g/kg. The two right columns represent rain affected data, where in the third column the observed specific humidity at a height of 6.5 km is lower than  0.5 g/kg, and in the last row it is larger than 2.7 g/kg. The grid here corresponds to hexagons with a diameter of  30 deg. Only those with a minimum of 15 observations inside them are shown.

 than in ECMWF high resolution analysis and GFS operational analysis, showing the different performance of  the analyses and re-analysis, with a smaller bias for the higher resolution analyses. On the other hand, for no rain and low specific humidity, the performance of the different analyses and re-analysis is similar.  Overall, the fact that the  bias is positive is an indication that models tend to be biased dry. This is in agreement with Hersbach et al. (2015),  who noticed a dry bias in ERA-Interim which was attributed to a problem in assimilating microwave  radiances affected by rain.

**6   Summary and discussion**

[revised manuscript text omitted]